# Hippocampus as a sorter and reverberatory integrator of sensory inputs

Masanori Nomoto [1,2,3], Emi Murayama[1,2,3], Shuntaro Ohno[1,2,3], Reiko Okubo-Suzuki[1,2,3], Shin-ichi Muramatsu [4,5] & Kaoru Inokuchi [1,2,3] ✉

The hippocampus must be capable of sorting and integrating multiple sensory inputs separately but simultaneously. However, it remains to be elucidated how the hippocampus executes these processes simultaneously during learning. Here we found that synchrony between conditioned stimulus (CS)-, unconditioned stimulus (US)- and future retrieval-responsible cells occurs in the CA1 during the reverberatory phase that emerges after sensory inputs have ceased, but not during CS and US inputs. Mutant mice lacking N-methyl-D-aspartate receptors (NRs) in CA3 showed a cued-fear memory impairment and a decrease in synchronized reverberatory activities between CS- and US-responsive CA1 cells. Optogenetic CA3 silencing at the reverberatory phase during learning impaired cued-fear memory. Thus, the hippocampus uses reverberatory activity to link CS and US inputs, and avoid crosstalk during sensory inputs.

The hippocampus, a center of multimodal convergence, is crucial for learning and memory of associative episodes[1]. There are two parallel pathways, the trisynaptic pathway important for one-trial contextual learning[2–4] and the monosynaptic pathway important for temporal association[5]. Among these pathways, the CA3 has a unique system, a recurrent circuit forming extensive interconnections within CA3 cells[6]. During sensory inputs in fear learning, the entorhinal-hippocampal inhibitory mechanism restricts the incorporation of unconditioned stimulus (US) input as a part of conditioned stimulus (CS) representation in the hippocampus[7]. On the other hand, aversive input propagates into the hippocampus and activates hippocampal neurons[8–10]. These findings imply a novel hippocampal function in which CS and US association occurs after the termination of sensory inputs. However, it remains to be elucidated how the hippocampus executes two distinct processes, sorting and integrating the CS and US inputs during memory encoding. Theoretical models have suggested that the CA3 recurrent circuit implemented with N-methyl-D-aspartate receptor (NR) function has a potential to generate reverberatory neuronal activities without input from external stimuli[11–13], and acts as an associator of temporally separated episodes by filling a temporal

gap between discontinuous events[14–18]. However, experimental studies have indicated that the CA3 recurrent circuit is not required for trace-type associative memory formation, which requires the ability to form the temporal association between events[5,19,20]. We hypothesized that, in the delayed conditioning in which the US is simultaneously paired with the preceding CS, the hippocampal network is programmed to process sensory information after the termination of sensory inputs and reverberatory activities, which could repeat the CS and the US representations, serve as an integrator to link neutral and aversive stimuli. We sought to determine whether and how reverberatory activities contribute to the CS-US association during conditioned learning. We defined the reverberatory activity as a trace activity of cells that is evoked by the sensory inputs and transiently lasts after the sensory inputs ceased.

## Results

### NRs and neuronal activities in the CA3 are important for cued-fear memory

We subjected mutant mice that specifically lack NRs in CA3 (CA3-NR1 KO mice), and thus are deficient in NR current and synaptic plasticity at

[1]Research Centre for Idling Brain Science, University of Toyama, Toyama 930–0194, Japan. [2]Department of Biochemistry, Graduate School of Medicine and Pharmaceutical Sciences, University of Toyama, Toyama 930–0194, Japan. [3]CREST, JST, University of Toyama, Toyama 930–0194, Japan. [4]Division of Neurology, Department of Medicine, Jichi Medical University, Tochigi 329–0498, Japan. [5]Center for Gene and Cell Therapy, The Institute of Medical Science, The University of Tokyo, Tokyo 108–8639, Japan. ✉e-mail: inokuchi@med.u-toyama.ac.jp

the recurrent CA3 synapses[21], to a delayed-type light-cued-fear conditioning (LFC) task[22] (Fig. 1a, b). CA3-NR1 KO mice and littermate controls were both subjected to pre-contextual habituation followed by training sessions to form an association between a light-cue CS and a footshock US with ten pairings. In the following test, CA3-NR1 KO mice exhibited impairment in long-term, but not short-term, cued-fear memory recall relative to littermate controls (Fig. 1c, d). By contrast, CA3-NR1 KO mice did not exhibit impairments in long-term contextual fear memory in the LFC task (Fig. 1e) or in an alternative contextual fear memory task, which consisted of context pre-exposure and immediate footshock sessions[23,24] (Supplementary Fig. 1). Our results testing

contextual fear memories in CA3-NR1 KO mice confirmed previous findings, which suggested that CA3 NRs are important for novel contextual representation but not familiarized context representation[2–4]. Furthermore, long-term memory recall of an auditory-cued-fear conditioning (AFC) task was impaired in CA3-NR1 KO mice relative to controls (Fig. 1f, g), indicating that CA3 NRs are important for long-term cued-fear memory.

Lentivirus (LV) encoding calcium/calmodulin-dependent protein kinase II (CaMKII)-FLEX-eArchT3.0-EYFP was bilaterally injected into the hippocampal CA3 of KA1::Cre heterozygous transgenic mice to specifically label and silence CA3 excitatory cells (Fig. 1h, i). One day after the training session, mice were subjected to test sessions during which a continuous laser (589 nm) was bilaterally delivered to the CA3, starting at the onset of the first CS in the LFC or AFC tasks (Fig. 1j). Mice with precise optical CA3 silencing exhibited impaired long-term cued-fear memory recall in both LFC and AFC tasks relative to the control (laser-OFF) group (Fig. 1k, l). These results indicated that NRs and neuronal activities in the CA3 are important for cued-fear memory, which is consistent with previous reports[8,25,26] indicating involvement of the hippocampus in cued-fear memory.

### The lack of CA3 NRs does not alter the CA1 ensemble structure responsive to sensory inputs

To investigate how the hippocampus processes CS and US information, and whether reverberatory activity emerges after termination of sensory stimuli in the cued-fear conditioning task, we monitored in vivo transient calcium (Ca2+) dynamics in hippocampal cells. CA3-NR1 KO mice and littermate controls were injected with an Adeno-Associated Virus (AAV) encoding CaMKII-G-CaMP7 and implanted with a micro-gradient index (GRIN) lens targeting the right CA1 (Fig. 2a). The same CA1 cells were tracked across LFC task sessions, including a 30 min rest session after the training, using an automated sorting system to extract the Ca2+ activity of each neuron, which was then normalized using z-scores (Supplementary Fig. 2).

By calculating and comparing mean z-scores as indicators of the responsiveness for each session (please refer to Methods), we sorted training CS (CS)-, training US (US)-, and long-term memory-test CS (Test-CS)-responsive subpopulations of cells that exhibited 2-fold higher responses to stimuli than in the baseline session on the training day before the corresponding CS or US (Fig. 2b–e). We did not detect structural differences of these CA1 subpopulations between CA3-NR1 KO mice and littermate controls (Fig. 2c–f). About half of the Test-CS-responsive cells had newly emerged and were not CS-responsive cells, indicating that this cell subpopulation that responded to the CS changed from the training session to the test session (Fig. 2g).

### CA3 NR-dependent reverberatory activity occurs in CA1 during cued fear learning

Notably, US input immediately and completely shut down the activity of CS-responsive and Test-CS-responsive cells (Fig. 3a, c). The Ca2+ activities of the Test-CS-responsive cells, but not the CS-responsive or US-responsive cells, during CS presentation in the test session were higher than those in the corresponding acclimation (Test-Acc) and inter-trial interval (Test-ITI) sessions (Fig. 2e). CS-responsive cells in the CA3-NR1 KO mice exhibited dramatically decreased Ca2+ activity relative to the control during the ITI, especially during the first 15 s after the CS-US presentation (ITI-1), and also during the 30 min rest session (Fig. 3a, Supplementary Fig. 3, and Supplementary movie 1). The activities of US-responsive cells were comparable between genotypes throughout all sessions (Fig. 3b). Ca2+ activity of the Test-CS-responsive cells emerged after CS-US presentation (Fig. 3c). CS-responsive and Test-CS-responsive cells in CA3-NR1 KO mice exhibited less activity during the ITI-1 than the control. Together, these studies indicated that in the CA1, ablation of CA3 NRs decreases reverberatory activities in current (CS) and future (Test-CS) CS-responsive cell

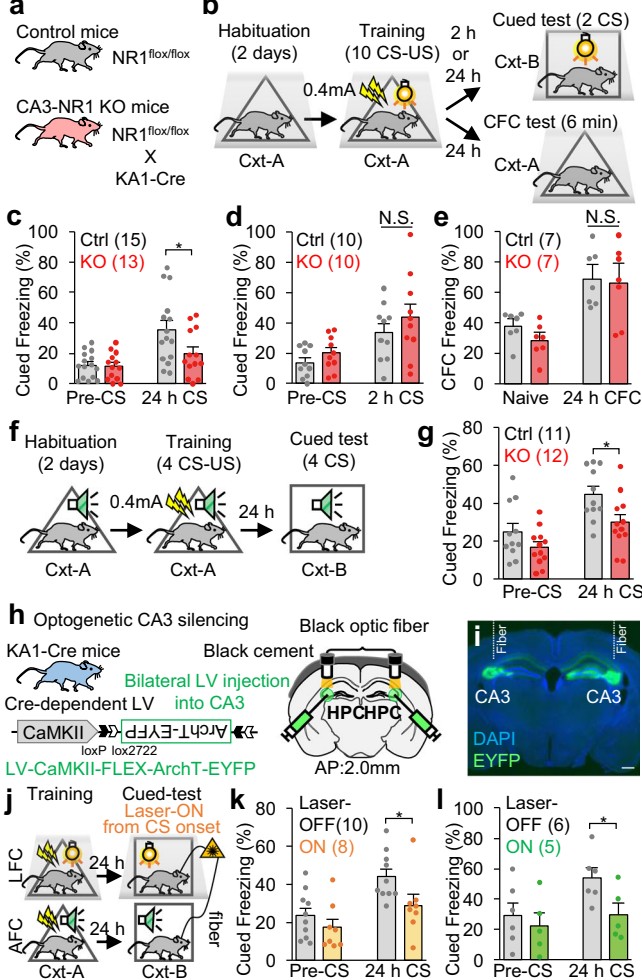

Fig. 1 | CA3 NRs and CA3 activity are important for cued-fear memory. **a** Animals used in this study. **b** Light-cued-fear conditioning (LFC) task. **c, d** Cued freezing levels during (**c**) 24 h long-term (two-tailed unpaired Student's t test, $P = 0.046$), and (**d**) 2 h short-term, memory tests. **e** Contextual freezing levels during a 24 h long-term memory test in the LFC task. **f** Auditory-cued-fear conditioning (AFC) task. **g** Cued freezing levels during the 24 h long-term test in AFC task (two-tailed unpaired Student's t test, $P = 0.02$). **h** Animal and virus vector used for optogenetic CA3 silencing. **i** Coronal section of the hippocampus with EYFP-expressing cells. All of the animals incorporated as data showed similar expression pattern. Scale bar, 500 μm. **j** Schedule for optogenetic experiment. **k, l** Cued freezing levels during the 24 h long-term memory test in the (**k**) LFC (two-tailed unpaired Student's t test, $P = 0.039$), and (**l**) AFC tasks (two-tailed unpaired Student's t test, $P = 0.04$). P values determined using an unpaired two-tailed t test (*$P < 0.05$) (**c–e, g, k, l**). Graphs represent the mean ± SEM, and circles within the graphs represent individual animals. Numbers in parentheses denote the number of mice in each group used for the study. Lightning bolt, footshock; Light bulb, light CS; Cxt, context; Speaker, tone CS; HPC, hippocampus; AP, anterior-posterior; N.S., not significant. Detailed statistics are shown in Supplementary Data 1.

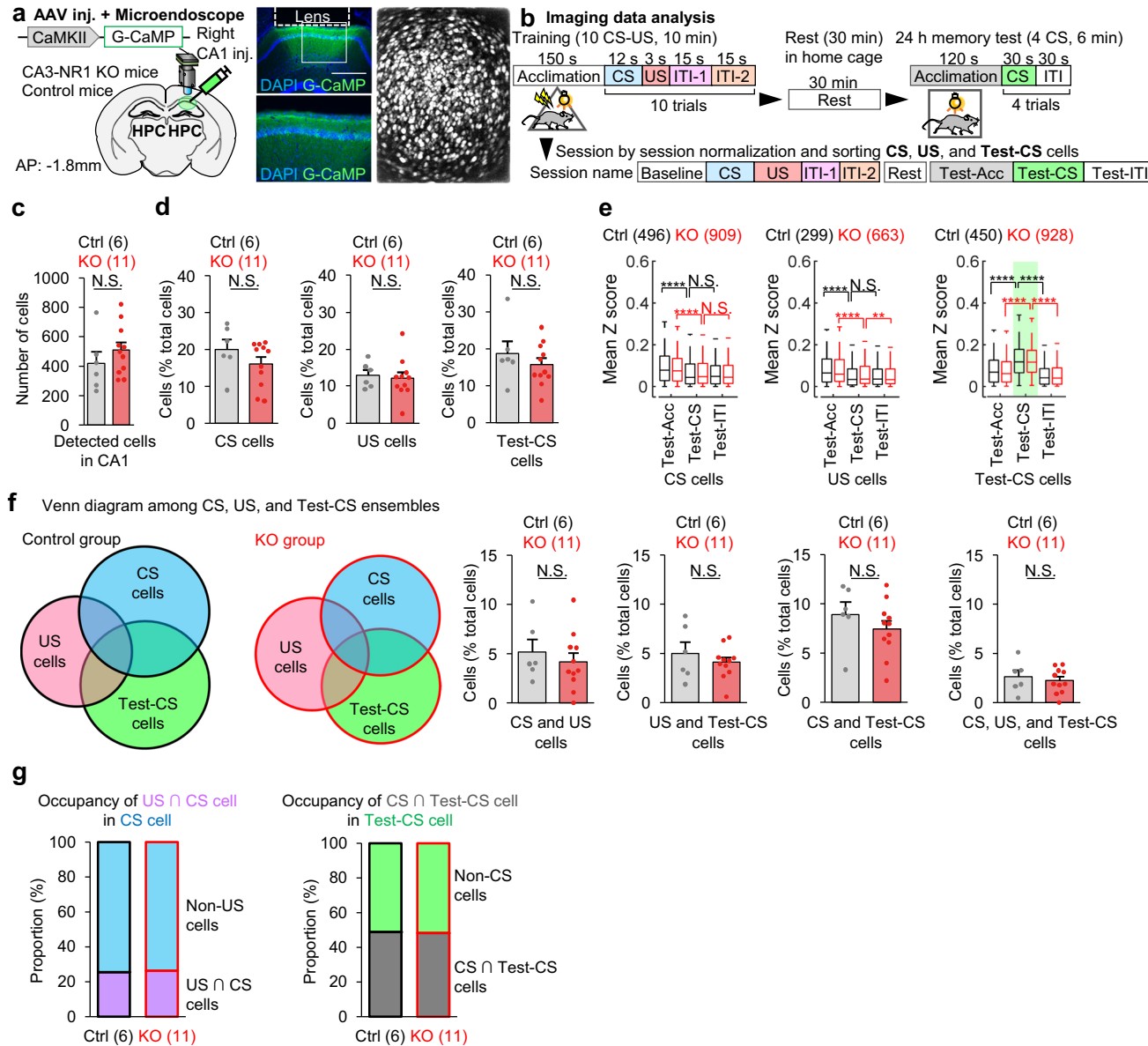

**Fig. 2 | CA3-NR1 KO mice exhibited normal CA1 ensemble structure. a** Left, experimental design. Right, coronal section of the hippocampus with G-CaMP expression, GRIN lens implantation, and stacked dF/F image acquired using a microendoscope over entire recording sessions of hippocampal imaging. Similar expression pattern was confirmed at least 5 times independently. Scale bar, 500 μm. **b** Imaging data analysis scheme. In each cell, Ca²⁺ data is classified into nine sessions, and the calculated mean z-score is considered to represent responsiveness and sorted into CS-, US-, and Test-CS-responsive subpopulations. **c** Columns comparing number of detected cells during CA1 imaging in control and CA3-NR1 KO mice. **d** Columns comparing percent of ensemble size in CS-, US-, and Test-CS-responsive subpopulations between control and KO mice. **e** Box plots comparing mean z-scores of long-term memory test sessions between genotypes in CS-, US-, and Test-CS-responsive subpopulations (two-tailed paired Wilcoxon signed-rank test: Ctrl CS cells Test-ACC to Test-CS, $P = 1.2E{-}06$; KO CS cells Test-ACC to Test-CS, $P = 5.4E{-}07$; Ctrl US cells Test-ACC to Test-CS, $P = 7.8E{-}07$; KO US

cells Test-ACC to Test-CS, $P = 7.3E{-}07$; KO US cells Test-CS to Test-ITI, $P = 0.004$; Ctrl Test-CS cells Test-ACC to Test-CS, $P = 3.1E{-}22$; KO Test-CS cells Test-ACC to Test-CS, $P = 2.0E{-}45$; Ctrl Test-CS cells Test-CS to Test-ITI, $P = 5.2E{-}36$; KO Test-CS cells Test-CS to Test-ITI, $P = 4.8E{-}68$). **f** Venn diagrams comparing and illustrating the overlapping and size of each ensemble in CA1. Columns comparing percent of overlapping ensemble sizes among CS-, US-, and Test-CS-responsive subpopulations between control and KO mice. **g** Left, occupancy of US ∩ CS-responsive cells in the CS-responsive cell population. Right, occupancy of CS-responsive cells in the Test-CS-responsive population. Numbers in parentheses denote the number of mice (**c, d, f**) or cells (**e**) in each group used for the study. *P* values were calculated using an unpaired two-tailed *t* test (**c, d, f**) or two-tailed Wilcoxon signed-rank test (**e**) (**$P < 0.01$, ****$P < 0.001$). N.S., not significant ($P > 0.05$). Box plots illustrate median, first, and third quantiles, and minimum and maximum values. Graphs represent means ± SEM, and circles in the graphs represent individual animals. Detailed statistics are shown in Supplementary Data 1.

ensembles, which emerge after termination of sensory stimuli, without affecting the ensemble structure.

The postulated advantage of the reverberatory activity is to prolong the time window that allows temporal coordination among cell ensembles to integrate stimuli, leading to associative memory formation[11–13]. To determine if synchronized activity among cell ensembles increases during the ITI-1, we counted the number of

pairwise synchronized Ca2⁺ activities within 500 ms (please refer to Methods) (Supplementary Fig. 4). CA3-NR1 KO mice exhibited significantly lower pairwise synchrony between newly generated Test-CS-responsive cells (Test-CS-specific cells) and (CS ∪ US)-responsive cells during the ITI-1 than the control group. Furthermore, triple synchrony between CS-, US-, and Test-CS-specific cells in CA3-NR1 KO mice was significantly lower than that of littermate controls only during the ITI-1

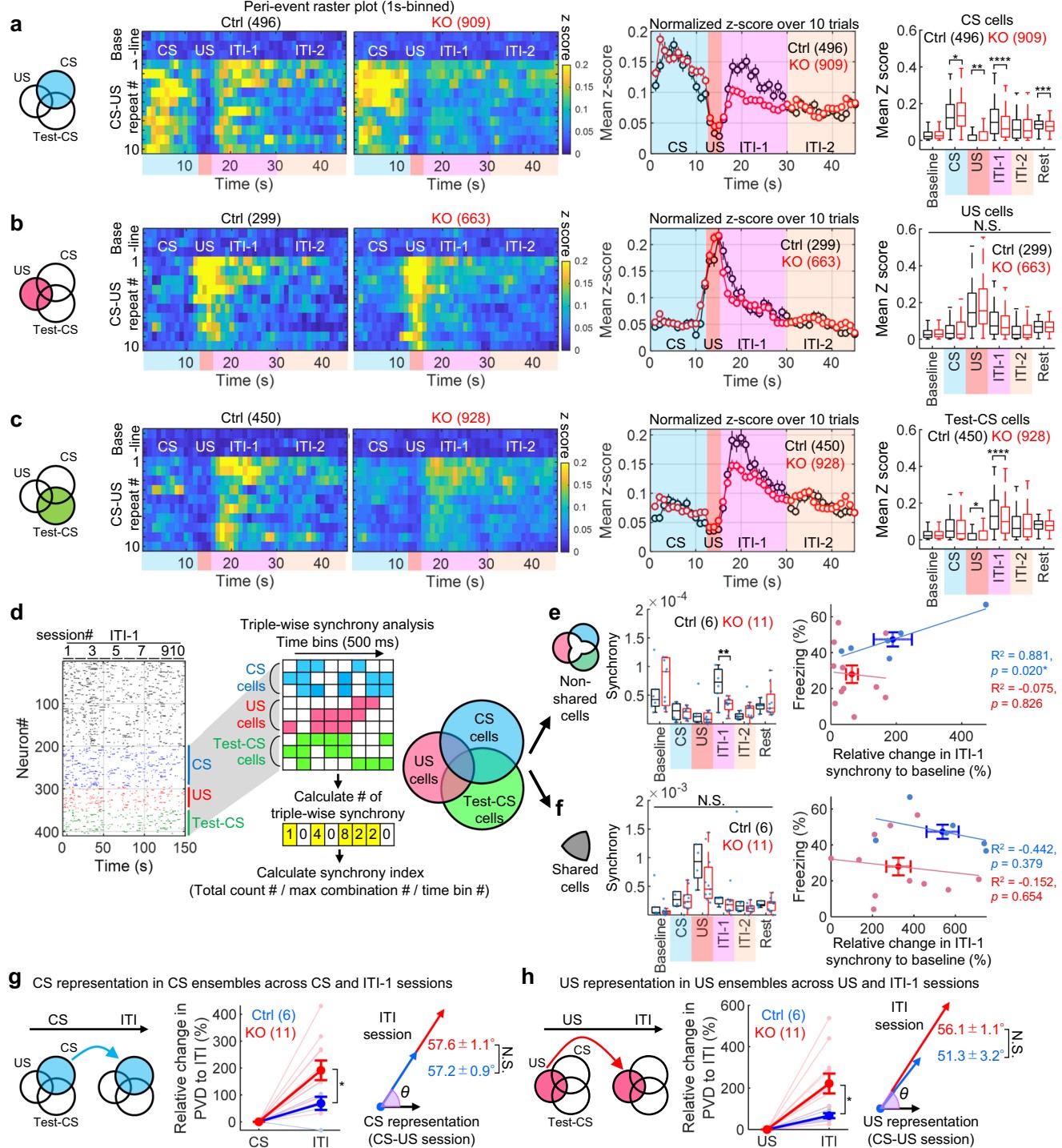

**Fig. 3 | CA3 NRs are involved in reverberatory and synchronized activity, but not sensory propagation, in CS- and Test-CS-responsive CA1 ensembles. a-c** Left, Venn diagrams representing (**a**) CS-, (**b**) US-, and (**c**) Test-CS-responsive ensembles. Peri-event raster plots during the training session in each subpopulation of littermate control and KO mice. Each short vertical tick represents a 1 s change of mean z-score across baseline and ten CS-US pairings. Ca2+ activities were aligned at the time that CS-US stimuli were delivered. The color code represents mean z-score. Middle, averaged z-score plots over ten CS-US pairings in each subpopulation. Right, box plots comparing mean z-scores between genotypes in each session (two-tailed unpaired Wilcoxon rank sum test for CS cells: CS, $P = 0.045$; US, $P = 0.004$; ITI-1, $P = 5.9E-10$; Rest, $P = 5.9E-10$; for Test-CS cells: US, $P = 0.011$; ITI-1, $P = 0.00005$). **d** Left, representative binarized raster plots of Ca2+ activity across ten ITI-1 sessions in control animals. Right, magnified raster plots focusing on CS-, US-, and Test-CS-responsive subpopulations and scheme for synchrony analysis. This analysis calculates synchrony by normalizing the number of synchronizations every 500 ms among the three subpopulations in each session. **e, f** Box plots comparing mean synchrony between genotypes in each session (two-tailed unpaired Student's t test: Non-shared cells ITI-1, $P = 0.0027$).

**g, h** Mahalanobis PVD and rotation between CS and ITI-1 sessions in the (**g**) CS-responsive ensemble (two-tailed unpaired Student's t test, CS-ITI PVD: $P = 0.035$) and between US and ITI-1 sessions in the (**h**) US-responsive ensemble (two-tailed unpaired Student's t test, US-ITI PVD: $P = 0.034$). Numbers in parentheses denote the number of (**a-c**) cells or (**e-h**) mice in each group used for the studies. $P$ values were determined using a (**a-c**) two-tailed Wilcoxon rank sum test (*$P < 0.05$, **$P < 0.01$, ***$P < 0.001$, ****$P < 0.0001$). N.S., not significant ($P > 0.05$), (**e-h**) Unpaired two-tailed t test (*$P < 0.05$, **$P < 0.01$). N.S., not significant ($P > 0.05$) or Pearson correlation (*$P < 0.05$). Box plots represent median, first, and third quantiles, and minimum and maximum values. Graphs represent means ± SEM. Detailed statistics are shown in Supplementary Data 1.

(Fig. 3d, e). Notably, the control group exhibited a positive correlation between the relative degree of animals' freezing on stimulus and the degree of ITI-1 synchrony. These characteristic features of synchrony were not observed in shared cells (Fig. 3f).

To further assess the representation similarity between CS and US cell ensembles across CS, US, and ITI-1 sessions, we calculated the Mahalanobis population vector distance (PVD)[27,28] following principal component analysis (PCA)-based dimension reduction[29] and rotation of multidimensional population vectors (one dimension per cell)[27] (Fig. 3g, h). CS or US ensembles in CA3-NR1 KO mice and littermate controls exhibited comparable rotation from CS or US to ITI (Fig. 3g, h). Both CS and US ensemble representations were more stable in littermate control mice than in CA3-NR1 KO mice across sessions, as demonstrated by small PVD changes from the stimulus to the ITI session. This suggests that reverberatory activities repeat CS and US representations. By contrast, ensemble activities exhibited increased variation across sessions in CA3-NR1 KO mice.

### CA3 NRs are important for CA1 and CA3 reverberation of CS, but not US inputs

The hippocampus processes multimodal information and contains a wide variety of cell types, such as place and head-direction cells[30], which could contribute to the observed reverberatory activity. Thus, we determined if the sensory stimuli alone (CS or US) triggers reverberatory activity in the hippocampal network using head-fixed mice. Mice operated for imaging in the CA1 (Fig. 2a) and the CA3 (see below) were additionally prepared in such a manner as to allow head fixation on a head-fixed apparatus consisting of a footshock grid and a light bulb via a holding bar with dental cement (Fig. 4a). After habituation for 4 days, mice were subjected to a training session in which they were exposed to the CS or the US alone using the same exposure time and interval as in the LFC task (Fig. 4b). The training CS- and US-responsive subpopulations were sorted using the same criteria (2-fold higher responses to stimuli). The CS-responsive cells in both the CA1 and CA3 of CA3-NR1 KO mice exhibited significantly lower Ca2[+] activity than the littermate controls during the ITI-1 and the ITI-2 (Fig. 4c, e, Supplementary Fig. 5, and Supplementary movie 2). By contrast, the US-responsive cells exhibited comparable activities in both genotypes (Fig. 4d, f). These findings, combined with the findings in freely moving mice, demonstrate that during sensory input, CA3 NRs are not important for direct propagation of CS and US information into the hippocampal CA3-CA1 network, but rather are crucial for reverberation of CS, but not US, representation in this network.

### CA3 NRs are important for reverberation of CS stimuli in freely moving condition

We also examined CA3 dynamics during the LFC task in freely moving conditions (Fig. 5a, b). There were no significant differences in the cell ensemble structure or in the Ca2[+] activities during test sessions between CA3-NR1 KO mice and littermate controls (Fig. 5c–f). Similar to the CA1, US input partially inhibited the Ca2[+] activities of CS- but not Test-CS-responsive cells. The CS-responsive cells in CA3-NR1 KO mice exhibited significantly lower Ca2[+] activities during the 9 s of the ITI-1 (6 to 14 s of 15 s in ITI-1, later 9 s) than the control (Fig. 6a). CS-, US-, and Test-CS-responsive cells in CA3-NR1 KO exhibited significantly lower Ca2[+] activities during the ITI-2 sessions than the control (Fig. 6a–c). Triple synchrony was comparable between CA3-NR1 KO and littermate control mice for the duration of LFC training sessions (Fig. 6d–f).

We further compared the synchrony rate (Hz) in ITI-1 that is an indicator of the frequency of synchronous events (Fig. 6g). Two-way ANOVA for the synchrony rate in ITI-1 revealed a significant effect of hippocampal region, but not genotype and hippocampal region vs. genotype interaction (Fig. 6g). Two-way ANOVA for the synchrony index in ITI-1 revealed a significant effect of interaction between hippocampal region vs. genotype (Fig. 6h). The CA1 synchrony index

during ITI-1 in littermate control was higher than the other groups (Fig. 6h). Thus, the deficit of CA3 NRs reduces the CA1 synchrony to the same level as CA3.

The ensemble representation analyses revealed that the CS ensemble representations, but not US, were more stable in littermate control mice than in CA3-NR1 KO mice across sessions, as demonstrated by both small PVD and ensemble rotation changes from the stimulus to the ITI session (Fig. 6i, j). These hippocampal CA3 and CA1 imaging results suggested that after the termination of sensory stimuli, the CA3-CA1 pathway acts as a reverberatory network of episodes in a CA3 NR-dependent manner.

### Reverberatory activity in CA3-CA1 pathway is crucial for cued fear memory

Finally, we determined if CA3 reverberatory activity is crucial for the association between the CS and US. KA1::Cre/CA3-NR1-KO mice were bilaterally injected in the CA3 with an AAV encoding chicken beta actin (CBA)-FLEX-ArchT-tdTomato to specifically label CA3 cells with ArchT-tdTomato. Wireless optogenetic LED (590 nm) cannulae were implanted bilaterally into the CA3 (Fig. 7a, b). CA3 neuronal activity was optogenetically silenced either during the ITI for 10 s immediately after CS-US presentation (Tr) or during a 10 min rest period after the training session (HC) in cued-fear conditioning using the same silencing intervals (please refer to Methods) (Fig. 7c). One day after the training session, mice were subjected to a cued-fear memory test followed by a contextual fear memory test at a 1 h interval (Fig. 7d, e). Consistent with the behavioral data in Fig. 1, CA3-NR1 KO mice exhibited impaired cued-fear memory and unchanged contextual memory recall compared with KA1::Cre mice (KA1 HC-ON vs KO HC-ON) (Fig. 7d, e). Importantly, mice that received silencing at the time of the early reverberatory phase (Tr-ON) exhibited significantly decreased cued-fear recall, but similar contextual fear memory in both genotypes compared with the group silenced after training (HC-ON). Taken together, these findings suggested that CA3 reverberatory activity is crucial for cued-fear memory encoding (Fig. 7f).

## Discussion

We detected time-limited and CA3 NR-dependent reverberatory activities that lead to synchronized activity among cell ensembles in the CA1. Our results suggest that the CA1 integrates episodes through the synchronized activity during the reverberation phase. The CA3 to CA1 network functions as a reverberatory and associative system of stimuli, in which the CA3 acts as a reverberator and the CA1 functions as both a reverberator and an integrator of episodes.

This prompts the question of why CS and US events must interact during the reverberatory phase. Simultaneous encoding of multiple stimuli in the brain neural network is limited by the capacity of cognition[31]. The hippocampus processes distinct valences, such as contextual and temporal episodes, with distinct cell subpopulations[32–35] and circuits[2–5]. The hippocampus could use reverberatory activity to avoid crosstalk during sensory inputs to store neutral and aversive information separately, and subsequently link the CS and US during the reverberatory phase. Indeed, a circuit mechanism in which CA1 activity is temporally regulated by EC input prevents crosstalk between CS and US stimuli during contextual CS and aversive US presentations[7]. Thus, the hippocampus functions as a sorter to encode CS and US independently, and subsequently as a reverberatory integrator to link CS and US.

CS-responsive cells occupied about half of Test-CS-responsive cells, that is, another half of the Test-CS cells emerged after the conditioning. The significant correlation between relative degree of animal freezing and triple synchrony of non-shared cells during ITI strongly suggests that, by synchronized activity, CS and US cells recruit and instruct newly generated Test-CS cells by synchronizing CS and US information (Supplementary Fig. 6). Thus, Test-CS cell population

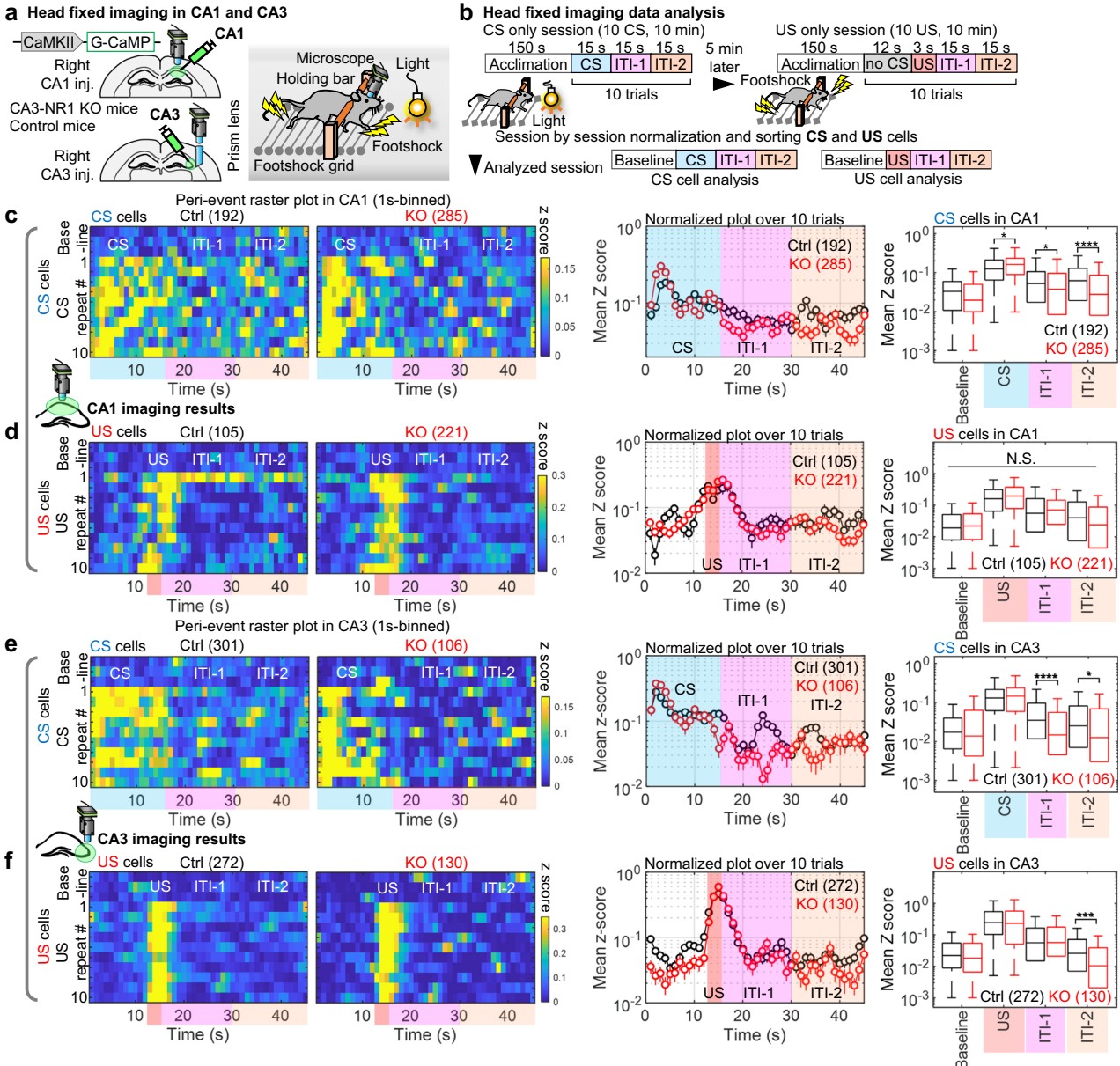

**Fig. 4 | CA3 NRs-dependent reverberation by single stimuli in the hippocampal network under head-fixed conditions. a** Left, experimental design. Right, schema for head-fixed imaging. The same footshock grid and light bulb were used in free-moving and head-fixed imaging experiments. **b** Imaging data analysis scheme. In each cell, Ca2+ data are classified into four sessions, and the calculated mean z-score, considered to represent responsiveness, is sorted into CS- and US-responsive subpopulations. **c-f** Left, peri-event raster plots during single (**c**) CS presentation session in CA1 (two-tailed unpaired Wilcoxon rank sum test for CA1 CS cells: CS, $P = 0.019$; ITI-1, $P = 0.013$; ITI-2, $P = 0.00001$), (**d**) US presentation session in CA1, (**e**) CS presentation session in CA3 (two-tailed unpaired Wilcoxon rank sum test for CA3 CS cells: ITI-1, $P = 0.00003$; ITI-2, $P = 0.021$), and (**f**) US presentation session in CA3 of (left) control and (right) KO mice (two-tailed unpaired Wilcoxon rank sum test for CA3 US cells: ITI-2, $P = 0.0003$). Each short

vertical tick represents a 1 s change of mean z-score across baseline, and CS or US presentations. Ca2+ activities are aligned at the time at which stimuli were delivered. The color code indicates mean z-score. Middle, averaged z-score plots over ten CS or US presentations. Right, box plots comparing mean z-scores between genotypes in each session. Numbers in parentheses denote the number of cells in each group used for the study. Data were acquired from Ctrl ($n = 2$ mice) and KO ($n = 2$ mice) groups for CA1 imaging, and from Ctrl ($n = 4$ mice) and KO ($n = 2$ mice) groups for CA3 imaging. $P$ values were determined using (**c-f**) a two-tailed Wilcoxon rank sum test (*$P < 0.05$, **$P < 0.01$, ***$P < 0.001$, ****$P < 0.0001$). N.S., not significant ($P > 0.05$). Box plots represent the median, first, and third quantiles, and minimum and maximum values. Graphs represent means ± SEM. Detailed statistics are shown in Supplementary Data 1.

serves as engram that drive the episodic recall of cued-fear memory. The hippocampus allocates CS and US events into distinct cell sub-populations (Figs. 2, 3, 5, and 6). In the amygdala, the CS-responsive cell ensemble begins to respond to US stimuli across repetitive CS-US presentations and eventually represents the US to encode cued-fear memory[27]. In addition, the amygdala exhibits greater overlapping in cell populations between training and retrieval sessions than the CA1[36]. Thus, the amygdala encodes CS-US association by altering the valence

representation of the cell ensemble. Our findings suggest that the hippocampus and amygdala adopt different strategies to encode distinct aspects of associative memory, episodic relation and direct linking, respectively.

Human studies reveal that in delayed conditioning in which the neutral cue (CS) is paired with the aversive stimulus (US), the hippocampus and amygdala act in parallel to associate the CS and US[37,38]. The hippocampus is crucial for declarative association between the CS and

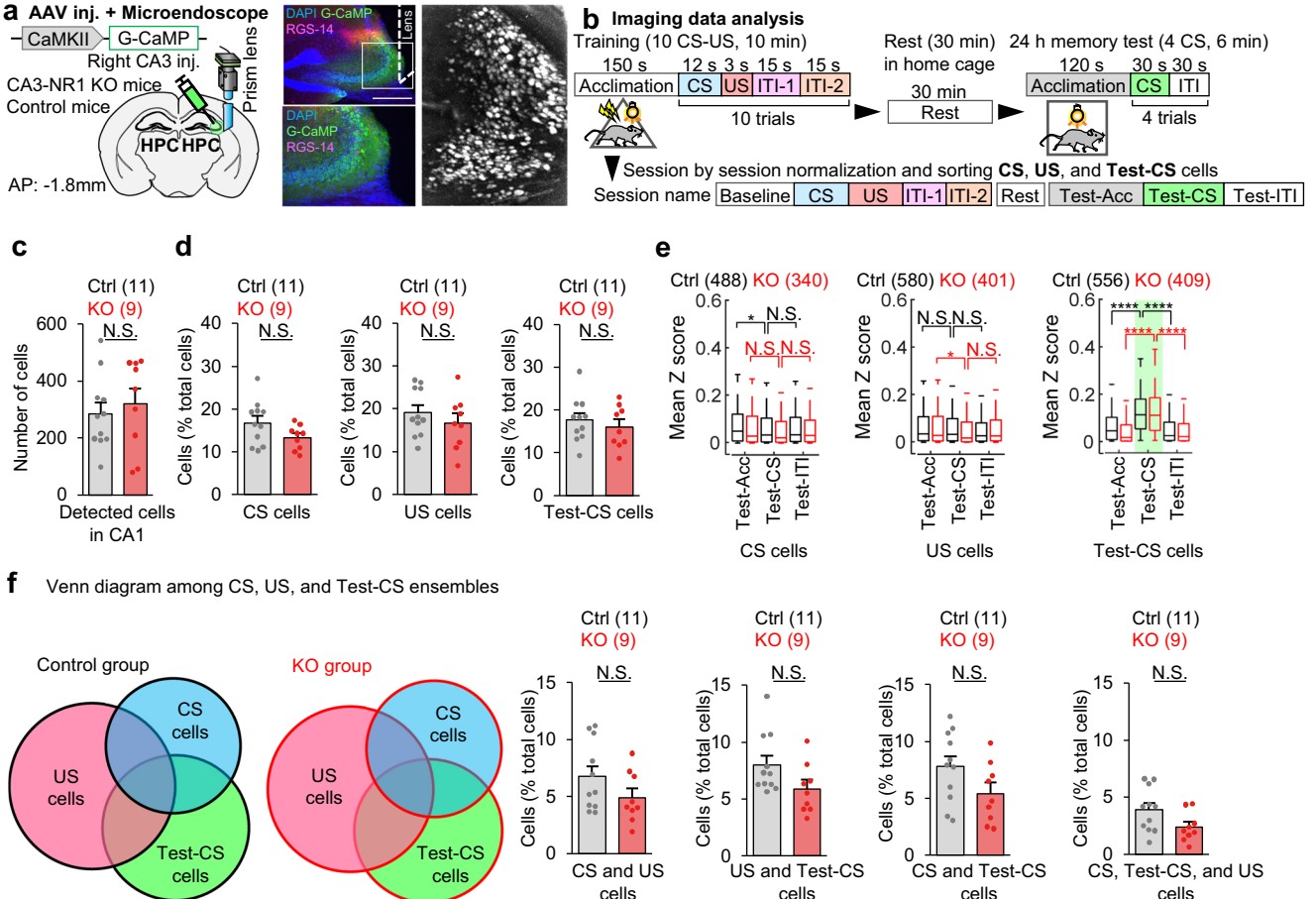

**Fig. 5 | CA3-NR1 KO mice exhibited normal CA3 ensemble structure. a** Left, experimental design. Right, coronal section of the hippocampus with G-CaMP-expressed cells (green) in CA3 and immunostained with anti-RGS-14 (red). RGS-14 is a marker for CA2 and stacked dF/F images acquired through the microendoscope over entire recording sessions of imaging in the hippocampus. Similar expression pattern was confirmed at least 5 times independently. Scale bar, 500 μm. **b** Imaging data analysis scheme. In each cell, Ca²⁺ data is classified into nine sessions, and the calculated mean z-score is considered to represent responsiveness and sorted into CS-, US-, and Test-CS-responsive subpopulations. **c** Columns comparing the number of detected cells during CA3 imaging in control and CA3-NR1 KO mice. **d** Columns comparing percent of ensemble sizes in CS-, US-, and Test-CS-responsive subpopulations between control and KO mice. **e** Box plots comparing mean z-scores of long-term memory test sessions between genotypes in CS-, US-, and Test-CS-responsive subpopulations (two-tailed paired Wilcoxon signed-rank

test: Ctrl CS cells Test-ACC to Test-CS, *P* = 0.012; KO US cells Test-ACC to Test-CS, *P* = 0.036; Ctrl Test-CS cells Test-ACC to Test-CS, *P* = 1.8E-31; KO Test-CS cells Test-ACC to Test-CS, *P* = 2.7E-28; Ctrl Test-CS cells Test-CS to Test-ITI, *P* = 1.9E-40; KO Test-CS cells Test-CS to Test-ITI, *P* = 2.5E-28). **f** Venn diagrams comparing and illustrating the overlapping and size of each ensemble in CA3. Columns comparing percent of overlapping ensemble sizes between CS-, US-, and Test-CS-responsive subpopulations between control and KO mice. Numbers in parentheses denote the number of (**c, d, f**) mice or (**e**) cells in each group used for the study. *P* values were determined using (**c, d, f**) an unpaired two-tailed *t* test or (**e**) a two-tailed Wilcoxon signed-rank test (*P < 0.05, ****P < 0.001). N.S., not significant (*P > 0.05). Box plots represent median, first, and third quantiles, and minimum and maximum values. Graphs show means ± SEM. In graphs, circles represent individual animals. Detailed statistics are shown in Supplementary Data 1.

US, while the amygdala is involved in automatic conditioned responses. In rodents, general theory indicates that the hippocampus is dispensable for delayed conditioning[39,40]. However, when conditioning stimuli are weaker and do not trigger the amygdala as robustly, hippocampal contribution to behavior becomes apparent[25,26,41,42]. We suggest that, similar to humans, the mouse hippocampus integrates the episodic relation between the CS and US, while the amygdala mediates direct associations between the CS and US (Supplementary Fig. 7).

Recently, Miyawaki and Mizuseki have reported that the inter-regional coactivation of cell ensemble in the hippocampus, amygdala, and prefrontal cortex occurs during the cued-fear memory conditioning, consolidation, and retrieval[43]. The inter-regional coactivation may contribute to the communication between the hippocampus and the amygdala, in which the episodic portion of the CS-US information is transported from the hippocampus to the amygdala.

Consistent with previous reports[2–4], NR deficiency in the CA3 or CA3 silencing during reverberation did not impair contextual fear memories after pre-contextual habituation (Figs. 1e, 7e, and Supplementary Fig. 1). These findings suggest that reverberation is required for association of novel episodes but not for association with pre-existing memories.

The slow kinetics of NRs are thought to be crucial for holding evoked excitation within the recurrent network[17,18]. Therefore, reverberatory activity is initially generated through the CA3 recurrent circuit in an NR-dependent manner. The entorhinal-hippocampal time-limited gate opens immediately after the sensory stimulus[7,44], which promotes propagation to the CA1 and initiates reverberation. Synchronized activities between CS- and US-responsive cells is stochastically regulated during the ITI, generating novel Test-CS-responsive cells.

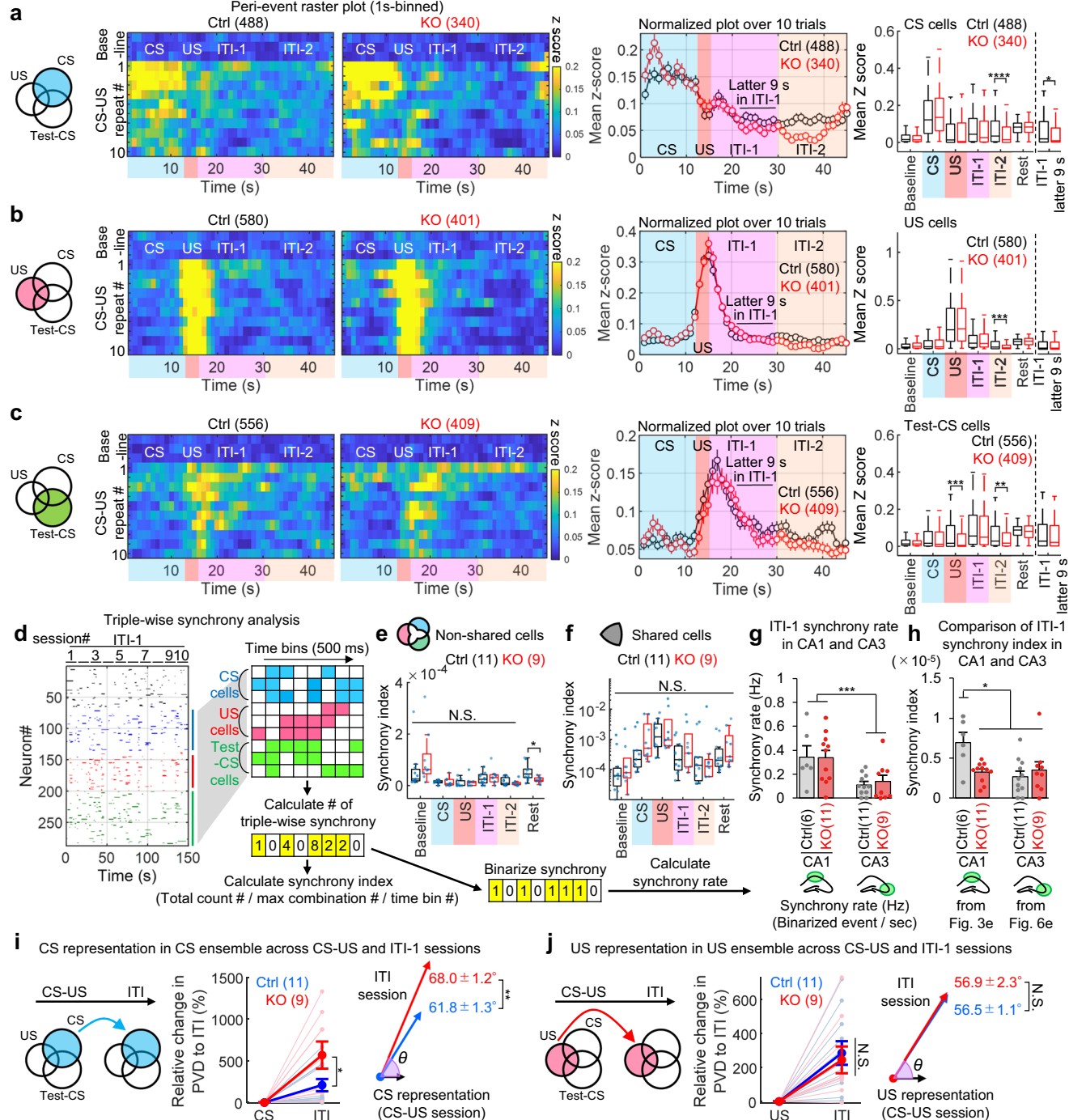

## Methods

### Mice

Male CA3-NR1 KO mice (C57BL/6 J background) and their floxed-NR1 littermate controls were used for behavioral, imaging, and optogenetic experiments. Male KA1::Cre mice were used for optogenetic experiments. CA3-NR1 KO mice were generated by crossing floxed-NR1 and KA1::Cre transgenic mice. Mice were maintained on a 12 h light-dark cycle at 24 °C ± 3 °C and 55% ± 5% humidity with standard laboratory diet and tap water ad libitum. All mice were aged 16–26 weeks at the time of behavioral experiments. All procedures involving the use of animals complied with the guidelines of the National Institutes of Health and were approved by the Animal Care and Use Committee of the University of Toyama (Approval numbers: A2019MED-35, A2022MED-7), Toyama, Japan.

### Viral constructs

For the in vivo Ca2+ imaging experiment, a recombinant Adeno-Associated Virus (AAV) vector encoding AAV9-CaMKII::G-CaMP7 (Titer: $9.4 \times 10^{12}$ vg/mL) after 40-fold dilution with phosphate-buffered saline (PBS) (T900; Takara Bio, Inc., Japan) was used[45]. For optogenetic silencing during the training session, AAV encoding AAV5-CBA-FLEX-ArchT-tdTomato, a gift from Edward Boyden (Titer: $1.3 \times 1013$ GC/mL) (Addgene viral prep # 28305-AAV5, USA) after 10-fold dilution with PBS was used. For optogenetic silencing during the test session, lentivirus (LV) encoding CaMKII-FLEX-eArchT3.0-EYFP, a gift from Karl Deisseroth (Titer: $5 \times 109$ IU mL−1) without dilution was used. The LV was prepared as described previously[32], according to the protocol developed by Karl Deisseroth.

**Fig. 6 | CA3-NR1 KO mice exhibited impaired reverberatory activity following CS-US presentation in CS- and Test-CS-responsive CA3 subpopulations under free-moving conditions. a–c** Left, venn diagrams representing (**a**) CS-, (**b**) US-, and (**c**) Test-CS-responsive ensembles. Peri-event raster plots during the training session in each subpopulation of littermate control and KO mice. Each short vertical tick represents a 1 s change of mean z-score across baseline and ten CS-US pairings. $Ca^{2+}$ activities were aligned at the time that CS-US stimuli were delivered. The color code represents mean z-score. Middle, averaged z-score plots over ten CS-US pairings in each subpopulation. Right, box plots comparing mean z-scores between genotypes in each session (two-tailed unpaired Wilcoxon rank sum test for CS cells: ITI-2, $P = 0.00002$; ITI-1 latter 9 s, $P = 0.012$; for US cells: ITI-2, $P = 0.00027$; for Test-CS cells: US, $P = 0.00052$; ITI-2, $P = 0.0043$). **d** Left, representative binarized raster plots of $Ca^{2+}$ activity across ten ITI-1 sessions in control animals. Right, magnified raster plots focusing on CS-, US-, and Test-CS-responsive subpopulations and scheme for synchrony analysis. This analysis calculates synchrony by normalizing the number of synchronizations in every 500 ms among three subpopulations in each session. Binarized synchrony data is used for calculation of synchrony rate. **e, f** Box plots comparing mean synchrony between genotypes in each session. **g** Synchrony rate between CA1 and CA3 and genotypes (Repeated measures two-way analysis of variance for brain region, $F_{(1,36)} = 14.40$, $P = 0.0006$). **h** Synchrony index during ITI-1 in CA1 and CA3 and genotypes (Repeated measures two-way analysis of variance for brain region × genotype interaction, $F_{(1,36)} = 9.441$, $P = 0.004$; two-tailed and adjusted Turkey-Kramer post hoc test for CA1 ctrl vs. CA1 KO, $P = 0.0221$; for CA1 ctrl vs. CA3 ctrl, $P = 0.0063$; for CA1 ctrl vs. CA3 KO, $P = 0.0472$). **i, j** Mahalanobis PVD and rotation (**i**) between CS and ITI-1 sessions in the CS-responsive ensemble, and (**j**) between US and ITI-1 sessions in the US-responsive ensemble (two-tailed Brunner-Munzel test for CS-ITI PVD, $P = 0.040$; two-tailed unpaired Student's t test for CS-ITI angle, $P = 0.003$). Numbers in parentheses denote the (**a–c**) number of cells or (**e–j**) mice in each group used for the study. $P$ values were determined using a (**a–c**) two-tailed Wilcoxon rank sum test ($*P < 0.05$, $**P < 0.01$, $***P < 0.001$, $****P < 0.0001$), (**e–f, i–j**) an unpaired two-tailed t test ($*P < 0.05$, $**P < 0.01$), a (**i**) two-tailed Brunner-Munzel test ($*P < 0.05$), or a two-way analysis of variance for (**g**) synchrony rate and (**h**) synchrony index with the adjusted and two-tailed Tukey–Kramer test ($*P < 0.05$, $**P < 0.01$, $***P < 0.001$). Significant effect of hippocampal region for **g**. N.S., not significant ($P > 0.05$). Box plots indicate median, first, and third quantiles, and minimum and maximum values. Graphs indicate means ± SEM. In graphs, circles represent individual animals. Detailed statistics are shown in Supplementary Data 1.

## Stereotaxic surgery for optogenetic and imaging studies

Stereotaxic surgery and optic fiber placement were conducted as described previously[32]. Prior to surgery, mice were anesthetized with intraperitoneal injection of a three-drug combination: 0.75 mg/kg medetomidine (Domitor; Nippon Zenyaku Kogyo Co., Ltd., Japan); 4.0 mg/kg midazolam (Fuji Pharma Co., Ltd., Japan); and 5.0 mg/kg butorphanol (Vetorphale; Meiji Seika Pharma Co., Ltd., Japan). After surgery, an intramuscular injection of 1.5 mg/kg atipamezole (Antisedan; Nippon Zenyaku Kogyo), a medetomidine antagonist, was administered to reverse sedation. Mice were placed on a stereotaxic apparatus (Narishige, Japan), and subsequently bilaterally injected with LV or AAV solution into the dorsal hippocampal CA3 (from bregma: +2.0 mm anteroposterior [AP]; ± 2.2 mm mediolateral [ML]; from dura: +1.8 mm dorsoventral [DV]). All virus injections were conducted using a 10 μL Hamilton syringe (80030; Hamilton, USA) fitted with a mineral oil-filled glass needle and wired to an automated motorized microinjector IMS-20 (Narishige). The glass injection tip was maintained before and after injection at the target coordinates for 5 min.

For the wired optogenetic experiment, mice were bilaterally injected with 500 nL LV solution at 100 nL min−1 into the CA3, and bilaterally implanted with guide cannulas targeting the CA3 (from bregma: −2.0 mm AP; ± 2.2 mm ML; from dura: +1.3 mm DV; C313GS-5/SPC, 22-gauge; Plastics One, USA). Dummy cannulas (C313IDCS-5/SPC, zero projection, Plastics One) were then inserted into guide cannulas to protect these guide cannula tubes from dust.

For the wireless optogenetic experiment, a wireless optogenetics system, Teleopto (Bio Research Center, Japan), was used[46]. Mice were bilaterally injected with 500 nL AAV solution at 100 nL min−1 into the CA3 and implanted with a dual-LED cannula (fiber diameter, 500 μm; fiber length, 3.3 mm; bilateral, 590 nm, 10 mW) targeting the CA3 (from bregma: −2.0 mm AP; ± 2.2 mm ML; from dura: +1.2 mm DV). Micro-screws were fixed near the bregma and lambda, and guide cannulas were fixed in position using dental cement (Provinice; Shofu, Inc., Japan) mixed with 5% carbon powder (484164; Sigma, USA). Mice were allowed to recover from surgery for 4 weeks in their home cages before behavioral experiments were initiated.

For the $Ca^{2+}$ imaging experiment, surgery was conducted as described previously with modifications[45]. Mice were unilaterally injected with 500 nL AAV9-CaMKII::G-CaMP7 at 100 nL min−1 into the right hippocampal CA1 (from bregma: −2.0 mm AP; + 1.4 mm ML; + 1.4 mm DV) or CA3 (from bregma: −2.0 mm AP; + 2.2 mm ML; from dura: +1.8 mm DV). After 1 week of recovery from AAV injection surgery, anesthetized mice were placed back onto a stereotactic apparatus to implant a gradient index (GRIN) lens into CA1 or CA3. A craniotomy (CA1, approximately 1.8 mm in diameter; CA3, approximately a $2.0 \times 2.0$ mm square) was performed centered over the injection site, and the neocortex and corpus callosum above the alveus overlying the dorsal hippocampal CA1 or CA3 were aspirated under constant irrigation with saline using a 26-gauge flat-blunted needle tip. Saline was applied to control bleeding. A cylindrical GRIN lens (diameter, 1.0 mm; length, 4 mm; Inscopix, USA) and prism GRIN lens (diameter, 1.0 mm; length, 4 mm with prism lens; Inscopix) were attached to the alveus and additionally squeezed 10–30 μm using handmade forceps attached to a manipulator (Narishige) for CA1 and CA3 imaging. Emulsified low-temperature bone wax was applied to seal the gaps between the GRIN lenses and the skull, and the lens was then anchored in place using dental cement mixed with 5% carbon powder (464164, Sigma) as described above. After the surgery, Ringer's solution (0.5 mL/mouse, i.p.; Otsuka, Japan) was injected, and atipamezole was administered as described above. Mice were maintained in individual cages after surgery. Three weeks after GRIN lens implantation, mice were anesthetized and placed back onto the stereotaxic apparatus to set a baseplate (Inscopix). A Gripper (Inscopix) holding a baseplate attached to a miniature microscope (nVista 3, Inscopix) was lowered over the implanted GRIN lens until visualization of clear vasculature was possible, indicating the optimum focal plane. Carbon-containing dental cement was then applied to fix the baseplate in position and preserve the optimal focal plane. Mice recovered from surgery in their home cages at least for 1 week before beginning behavioral imaging experiments.

For CA1 and CA3 head-fixed imaging experiments, mice that had undergone the surgery until the step of baseplate setting were anesthetized and placed back onto a stereotaxic apparatus, and mice were removably attached to a holding bar with a dental cement. The cut tips of a PCR tube held with ear bar for stereotaxic (Narishige) bilaterally were moved closer to the face between the eye and ear, and then the tips were fixed with dental cement, enabling the stereotaxic device to hold the mouse head on the apparatus.

## Behavioral analysis

All mice were numbered and randomly assigned to each experimental group before the experiments, with the exception of an imaging experiment. All behavioral experiments were performed and analyzed by an investigator blinded to experimental conditions with the exception of imaging experiment. For all behavioral procedures, animals in their home cages were moved on a rack to a resting room next to the behavioral testing room and left undisturbed for at least 30 min before each behavioral experiment. All behavioral chambers were

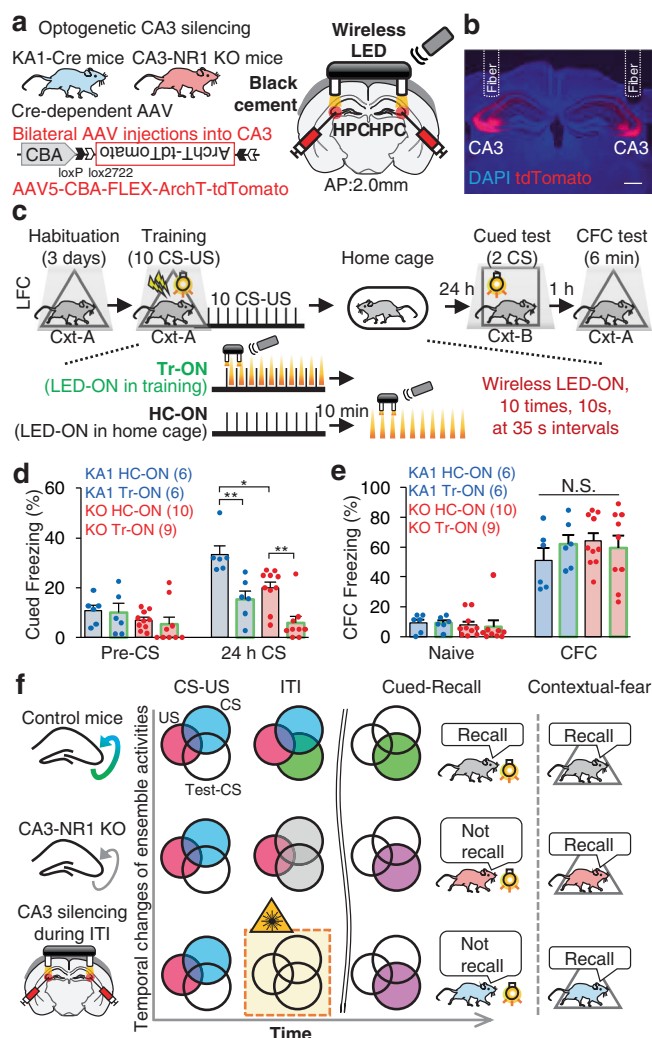

**Fig. 7 | CA3-CA1 pathway activity after termination of sensory stimuli is crucial for cued-fear memory encoding but not for contextual fear memory.**
**a** Experimental design. **b** Coronal section of the hippocampus with tdTomato expression and fiber implantation targeting CA3. All of the animals incorporated as data showed similar expression pattern. Scale bar, 500 μm. **c** Scheme for optogenetic manipulation. After the habituation session, mice were subjected to ten CS-US pairings in a training session. The CA3-CA1 pathway was silenced ten times for 10 s at 35 s intervals, during either the ITI phase following CS-US presentation (Tr-ON) or during resting in the home cage 10 min after the training session (HC-ON). On the next day, mice were tested for cued freezing and contextual freezing. **d** Cued freezing levels during the 24 h long-term memory test. Graphs represent means ± SEM, with circles indicating individual animals (Repeated measures two-way analysis of variance for genotype, $F_{(1,30)} = 13.996$, $P = 8.7\text{E-}4$; for opto, $F_{(1,30)} = 29.527$, $P = 9.5\text{E-}6$; for genotype × opto interaction, $F_{(1,30)} = 0.430$, $P = 0.517$; two-tailed and adjusted Turkey-Kramer post hoc test for KA1 HC-ON vs. KA1 Tr-ON, $P = 0.0031$; for KO HC-ON vs. KO Tr-ON, $P = 0.0035$; for KA1 HC-ON vs. KO HC-ON, $P = 0.0161$).
**e** Contextual freezing levels during the 24 h long-term memory test. Graphs represent means ± SEM, with circles indicating individual animals. $P$ values were determined using a (**d-e**) two-way ANOVA with the adjusted two-tailed Tukey–Kramer test (*$P < 0.05$, **$P < 0.01$). N.S., not significant ($P > 0.05$).
**f** Summarized scheme of imaging and behavioral results in the study. Venn diagrams of temporal changes in ensemble activities corresponding to CS, US, and Test-CS. Filled circles with color indicate the activated ensemble in each behavioral session of experimental groups. The CA3-NR1 KO group exhibited less activity in the CS- and Test-CS-responsive ensembles during ITI and failed cued-fear memory recalls. Numbers in parentheses denote the number of mice in each group used for the study. Lightning bolt, footshock; Light bulb, light CS; Cxt, context; HPC, hippocampus; AP, anterior-posterior; N.S., not significant. Detailed statistics are shown in Supplementary Data 1.

cleaned after each behavioral session. After all the optogenetic experiments were completed, the injection sites were histologically verified. Data were excluded from behavior analyses if the animals exhibited abnormal behavior after surgery, the target area was missed, or the bilateral expression of the virus was inadequate. All behavioral sessions were conducted using a video tracking system (Muromachi Kikai, Japan) to measure the freezing of mice. All sessions were recorded using Bandicam software (Bandisoft, Korea) or AG-desktop recorder software (T. Ishii, Japan). The cumulative duration (s) spent in the complete absence of movement, except for respiration, was considered to be the freezing duration. Automated scoring of the freezing response was initiated after 1 s of persistent freezing behavior, and the freezing data of optogenetic and imaging experiments were manually reanalyzed by other non-behavioral operators (E.M. and R.O.S.) in blinded condition with the same criteria, to exclude the effect of optogenetic device (optic fiber for wired optogenetics or Teleopt battery for wireless optogenetics) and calcium imaging device attachments on automated animal tracking.

## Light fear conditioning (LFC) task

LFC was conducted under dim light (approximately 2 lx) conditions as described previously with some modifications[22]. Two distinct contexts were used for LFC habituation, training, and testing sessions. For habituation, training, and contextual fear test sessions, a triangle-type chamber (context A: Cxt-A) with black stripe patterns was used. This chamber was a triangular prism (one side × height: 180 × 250 mm), with a transparent acrylic board for the front wall, black stripe-patterned side walls with an 8 W white light bulb, and a floor made from 26 stainless steel rods. For cued-fear test sessions, a quadrangular prism chamber (width × depth × height: 190 × 180 × 420 mm, respectively) was used (context B: Cxt-B), with a transparent acrylic front board and white side walls with an 8 W white light bulb and an asperity white floor. Before each test session, the white floor was scented with 0.25% benzaldehyde water. For the habituation session, mice were allowed to explore the Ctx-A apparatus for 6 min per day for 2 days (nonoperated, wired optogenetic, and imaging experiments) or for 3 days (wireless optogenetic experiment), and were then returned to their home cages.

For the training session 1 day after the habituation session, mice were conditioned in Cxt-A by ten pairings of the light-conditioned stimulus (CS) for 15 s with the unconditioned stimulus (US) (3 s footshock at the end of CS presentation, 0.4 mA) at 30 s intervals after a 150 s acclimation time. For the light-cued-fear memory test session, which was conducted 2 and 24 h after conditioning, different experimental mice were placed in Cxt-B for 120 s and then received two presentations of CS for 30 s at intervals of 30 s. For the contextual fear memory test session, 24 h after conditioning, different experimental mice were placed in Cxt-A for 360 s (non-operated animal experiments), while the same mice being tested for cued-fear memory were placed in Cxt-A for 360 s 1 hour after the light-cued-fear memory test (wireless optogenetic experiment).

## Auditory fear conditioning (AFC) task

AFC was conducted under normal light conditions as described previously with some modifications[47]. Two distinct contexts, described above, were used for AFC habituation, training, and testing sessions. For habituation and training sessions, a triangle-type chamber, Cxt-A, was used. This chamber had a transparent acrylic board for the front wall, black stripe-patterned side walls with a speaker, and a floor made from 26 stainless steel rods. For cued-fear test sessions, a quadrangular prism chamber, Cxt-B, was used. This chamber had a transparent acrylic front board and white side walls with a speaker and an asperity white floor.

For the 2 day habituation session, mice received four tone CS presentations (CS: 30 s at intervals of 30 s, 7 kHz, and 75 dB) after 120 s

of exposure to Cxt-A, and then were returned to their home cages. For training sessions 1 day after the habituation session, mice were conditioned in Cxt-A by four pairings of the tone CS for 30 s with the US (1 s footshock at the end of CS presentation, 0.4 mA) at 30 s intervals after a 120 s acclimation time. For the tone-cued-fear memory test session, 24 h after conditioning, mice were placed in Cxt-B for 120 s and then subjected to four CS presentations for 30 s at intervals of 30 s.

## Pre-exposure facilitated contextual fear conditioning (pre-exposure facilitated CFC) task

Pre-exposure-facilitated CFC was conducted as described previously with some modifications[24]. A quadrangular prism chamber, Cxt-B, with a footshock grid as described above was used. On day 1, mice were placed in Cxt-B for 6 min and then were returned to their home cages. On day 2, mice were placed again in Cxt-B and immediately given the US (1 s footshock); kept for 10 s in the context; and then returned to their home cage. On day 3, to assess contextual freezing, mice were placed in Cxt-B again for 360 s.

## Optogenetic experiments

Wired and wireless optogenetic experiments were conducted as described previously with some modifications[32]. For the wired optogenetic experiment, on the recall session, mice were anesthetized with 3% isoflurane for placement of the optical fiber units, and dummy cannulae were removed from the guide cannulae. The black-stained optical fiber unit, comprising a plastic cannula body, was a two-branch-type unit with a black-stained optic fiber diameter of 0.250 mm (COME2-DF2-250; Lucir, Japan). The optical fiber unit was inserted into the guide cannulae, and the guide cannulae and the optical fiber unit were tightly connected with the optical fiber caps (303/OFC, Plastics One). The tip of the optical fiber was targeted slightly above the hippocampal CA3 (from bregma: −2.0 mm AP, ± 2.2 mm ML; from dura: + 1.2 mm DV). Mice attached with an optical fiber were then returned to their home cages and left individually at least for 1 h before beginning the cued-fear test session. Immediately before beginning the test session, mice were moved to the experimental room, and the fiber unit connected to the mouse was attached to an optical swivel (COME2-UFC, Lucir), which was connected to a laser (200 mW, 589 nm, COME-LY589/200; Lucir) via a main optical fiber. The delivery of light pulses was controlled by a custom-made schedule stimulator with OpenEx Software Suite (RX8-2, Tucker Davis Technologies, USA) in synchronized mode with a behavioral video tracking system (Muromachi Kikai). During both LFC and AFC test sessions, optical illumination (continuous 589 nm light, approximately 5 mW output from the fiber tip) was delivered to the CA3 concurrently with the onset of the first CS in both LFC or AFC tasks, and maintained until the end of the test sessions. Mice were then returned to their home cages individually, and then the attached optic fiber was removed from the mice after the anesthesia. For post-hoc analysis, mice were deeply anesthetized with a mixed anesthesia solution as described above, and perfused transcardially with 4% paraformaldehyde in PBS (pH 7.4), followed by immunohistochemical analysis to confirm virus vector infection.

For the wireless optogenetic experiment, to allow habituation to the 2-gram battery units (Bio Research Center), attachment and removal of the battery units was initiated from the habituation session by anesthetizing mice with 3% isoflurane before and after each behavioral session, respectively. The battery unit was attached to the implanted Teleopt LED device above the head, and mice were then returned to their home cages and left individually for at least 1 h until initiating behavioral sessions. For habituation to Cxt-A, mice attached to battery units were placed in Cxt-A for 10 min per day for 3 days. The time for pre-context habituation in the wireless optogenetic experiment was extended compared to the non-operated and wired optogenetic experiments to get mice attached with the battery well

habituated to the chamber, because the battery's width was a little bit bigger than the widths of parts of optogenetic guide cannula and mice head. After 3 days of habituation sessions, mice were subjected to ten CS-US pairings of LFC in the training session. During the LFC training session, wireless optical illumination (590 nm continuous light, approximately 10 mW output from the fiber tip) was delivered to the CA3 region ten times for 10 s at 35 s intervals, during either the inter-trial interval phase following CS-US presentation (Tr-ON group) or during resting in the home cage 10 min after the training session (HC-ON group). Mice were then returned to their home cages individually, and the battery unit was removed under anesthesia. The delivery of light pulses was controlled by a custom-made schedule stimulator system as described above. One day after the training session, mice were attached to the battery unit and then subjected to a cued-fear memory test followed by a contextual fear memory test at 1 h intervals. For post-hoc analysis, mice were deeply anesthetized with the mixed anesthesia solution described above and perfused transcardially with 4% paraformaldehyde in PBS (pH 7.4), followed by immunohisto-chemical analysis to confirm virus vector infection.

## In vivo Ca2+ imaging data acquisition in freely moving and head-fixed animals

Attachment and removal of a microendoscope was performed under 3% isoflurane anesthesia before and after each behavioral experiment. Mice attached to the microendoscope were returned to their home cages to recover for at least 30 min before and after the behavioral session. For the freely moving imaging experiment, mice were habituated to the endomicroscope attachment for 10 min per day for 3 days in their home cages before beginning the LFC behavioral study. During the 2 day habituation session, mice were also attached to the microendoscope. In both habituation sessions using the home cage and behavioral context, calcium imaging was performed, but acquired data were not analyzed. Subsequently, actual imaging began from the LFC training session to the test sessions. For freely moving CA1 Ca2+ imaging, mice were subjected to training, 30 min resting after training, the 2 h short-term memory (STM) test, and then the 24 h long-term memory (LTM) test using the same LFC protocol. During the resting session, imaging was performed for 30 min. During the STM and LTM test sessions, after a 120 s acclimation session, mice were subjected to four CS presentations for 30 s at intervals of 30 s. However, STM session data were not analyzed because they were not important for the conclusion in this study. For freely moving CA3 Ca2+ imaging, mice were subjected to training, allowed to rest, and the 24 h LTM test was conducted using the same LFC protocol. During the resting session, imaging was performed for 30 min. During the LTM test sessions, after a 120 s acclimation session, mice were subjected to four CS presentations for 30 s at intervals of 30 s.

For the head-fixed imaging experiment, mice were habituated to endomicroscope attachment, and the fixation to the head-fixed apparatus comprised a footshock grid and an 8 W white light bulb, covered with a hemi-square paper box for 10 min per day for 4 days before beginning the head-fixed experiment. During habituation sessions, calcium imaging was performed, but acquired data were not analyzed. The footshock grid and light bulb were identical to the devices used for the freely moving LFC paradigm, and were measured with behavioral software (Muromachi Kikai). On the next day, mice were subjected to a CS (15 s constant light) presentation session and then a US (3 s footshock, 0.4 mA) presentation at intervals of 1 h. For the CS presentation session, after 150 s acclimation, mice were subjected to ten CSs for 15 s at 30 s intervals. For the US presentation session, after a 162 s acclimation time, mice were subjected to ten USs for 3 s at 42 s intervals. The durations of CS and US presentation sessions were 10 min total. In both freely moving and head-fixed imaging experiments, Ca2+ imaging was performed under dim light (approximately 2 lx) conditions, and the onset of behavioral and imaging

systems was synchronized using the OpenEx Software Suite (RX8-2, Tucker Davis Technologies). Ca$^2+$ signals produced from G-CaMP7 protein expressed in CA3 and CA1 excitatory neurons were captured at 20 Hz with nVista acquisition software (Inscopix) at the optimal gain and power of nVista LED. Ca$^2+$ imaging movie recordings of all behavioral sessions were then extracted from the nVista Data acquisition (DAQ) box (Inscopix).

For post-hoc analysis, mice were deeply anesthetized with the mixed anesthesia solution described above and perfused transcardially with 4% paraformaldehyde in PBS (pH 7.4), followed by immunohistochemical analysis to confirm virus vector infection.

### In vivo Ca2$^+$ imaging data processing and analysis

In both freely moving and head-fixed imaging experiments, only completely motion-corrected data was used for subsequent analysis. Data with an inadequate frame or that could not be corrected were excluded from analysis. Using Inscopix data processing software (IDPS, Inscopix) to create a full movie, recorded raw movies were temporally concatenated, spatially down-sampled (2×) and cropped, and then corrected for motion artifacts against a reference frame. A reference frame showing clear blood vessels as landmarks was chosen, and other frames were then aligned to the reference frame. Further motion correction was performed using Inscopix Mosaic software (Mosaic, Inscopix) as described previously[45,48]. The corrected full movie was then temporally divided into individual behavioral sessions using Inscopix Mosaic software. Subsequently, each individual session movie was low bandpass-filtered to reduce noise using Fiji software (NIH, USA) as described previously (see Supplementary Fig. 2). The change of fluorescence signal intensity (ΔF/F) for each behavioral session was subsequently calculated using Inscopix Mosaic software according to the formula ΔF/F = (F − Fm)/Fm, where F represents the fluorescence of each frame and Fm is the mean fluorescence for the entire session movie. Subsequently, movies representing each session were re-concatenated to generate full movies, including all sessions in the ΔF/F format. Finally, cells were identified using an automatic sorting system, and HOTARU and Ca$^2+$ signals of the detected cells over time were extracted in a (Ď; time × neuron) matrix format, as described previously.

Subsequent data processing and analysis were performed using a custom-made MATLAB code. Ca$^2+$ signals were subjected to high-pass filtering (> 0.01 Hz threshold) to remove low-frequency fluctuations and background noise in each Ca$^2+$ cell signal, in which negative values were replaced with "0". Using the filtered Ca$^2+$ signal, the z-scores of behavioral sessions (training, resting, and LTM test sessions) were separately calculated to normalize and detect Ca$^2+$ activities by thresholding (> 3 Standard Deviations from the ΔF/F signal) at the local maxima of the ΔF/F signal[48]. Then, to calculate the responsiveness of the cells to each behavioral event, z-scored Ca$^2+$ activity was temporally sorted into nine behavioral events consisting of training acclimation (baseline), training CS (CS), training US (US), inter-trial interval 1 (ITI-1; -0−15 s after US-CS), ITI-2 (-16−30 s after CS-US) sessions in the training session, resting (Rest) session, test acclimation (Test-Acc), CS (Test-CS), and ITI (Test-ITI) in LTM test sessions. The mean Ca$^2+$ activities corresponding to behavioral events were calculated, and then divided by each cell baseline event to index responsiveness across behavioral events. Afterwards, CS-, US-, and Test-CS-responsive subpopulations with 2× greater responsiveness to stimuli than that of the baseline event were sorted. Ca$^2+$ activities of these subpopulations were tracked across the LFC paradigm to calculate the mean Ca$^2+$ activities of nine behavioral sessions and 1 s average Ca$^2+$ activities for subsequent analyses, by which Ca$^2+$ activities between genotypes were compared.

For functional synchrony analysis, to detect the Ca2$^+$ event, the z-scores in CS-, US-, and Test-CS-responsive subpopulations as described above were binarized by thresholding (> 3 Standard Deviations from the ΔF/F signal) at the local maxima of the ΔF/F signal, and then were temporally down-sampled from 20 to 2 Hz data (500 ms binning). Subsequently, the functional synchrony, consisting of the number of synchronized activities among neurons of the subpopulations in each 500 ms time window, was calculated and normalized as the functional synchrony in each behavioral session. The Eqs. (1) and (2) are used for the calculations of pairwise synchrony index, and triple-wise synchrony index, respectively.

$$\text{Pairwise synchrony index} = \frac{1}{T} \frac{\sum_{t=1}^{T} n_{SessionA}(t) \cdot n_{SessionB}(t)}{N_{SessionA} \cdot N_{SessionB}} \quad (1)$$

$$\text{Triple} - \text{wise synchrony index} = \frac{1}{T} \frac{\sum_{t=1}^{T} n_{SessionA}(t) \cdot n_{SessionB}(t) \cdot n_{SessionC}(t)}{N_{SessionA} \cdot N_{SessionB} \cdot N_{SessionC}}$$
$$(2)$$

Where $n_{SessionA}(t)$ ($n_{SessionB}(t)$, $n_{SessionC}(t)$) is the number of Session A (Session B, Session C) cells that were active in the time bin $t$; $N_{SessionA}$ ($N_{SessionB}$, $N_{SessionC}$) is the total number of Session A (Session B, Session C) cells; and $T$ is the total number of time bins for each session.

When the correlation between cued freezing and functional synchrony during ITI-1 was calculated, the variable, the change in ITI-1 synchrony relative to the baseline session in each cell, was calculated and used for correlation analyses.

When the synchrony rate was calculated, the 1s-binarized synchronous activity during ITI-1 was divided by the 150 (150 s) to calculate the frequency of synchrony rate (Hz).

### Population vector analyses

Calculations for the Mahalanobis population vector distance (PVD) and population vector rotation were conducted as described previously with some modifications. For Mahalanobis population vector distance, the 1 s-averaged z-scores in CS- and US- responsive subpopulations described above were used after principal component analysis (PCA)-based dimension reduction[29]. Since Mahalanobis distance does not work well because of the curse of dimensionality when the number of cells/dimensions (p) is greater than the number of available samples (n), (p > n), PCA was used to reduce the number of cells/dimensions in the data sets The data sets of 1 s-averaged z-scores in subpopulations are reduced into a lower dimension, and subsequently, the top three PCA scores (PC1, PC2, and PC3) are used to calculate the Mahalanobis population vector distance via PCA to quantify the similarity of two sets of neuronal representations between the CS or US session and ITI-1 sessions in CS and US ensembles, respectively. We defined a group of 3-dimensional activity vectors, $x$, for each behavioral session (CS, US, or ITI-1) and calculated the PVD between the two representations. For example, Eq. (3) is used for the calculation of the Mahalanobis PVD ($M$) between sets of CS- and ITI-1-evoked ensemble activity patterns in the CS ensemble.

$$M^2 = (x - \mu)^T \Sigma^{-1} (x - \mu) \quad (3)$$

where $x$ and $\mu$ are the individual and mean population vectors for the ITI-1 and CS ensemble activities, respectively, and $x^T$ and $\mu^T$ are their transposes. The Mahalanobis distance accounts for differences in the means of the two sets of ensemble activities as well as their co-variances. The average PVD over all points $x$ in both sets of ensemble activities was calculated. To analyse the CS-ITI-1 and US-ITI-1 PVDs during the ten CS-US presentations in the CS and US ensembles, respectively, the top three PC scores calculated from the 1 s-averaged z-score data set are used, and the scores sorted by the CS, US, and ITI-1

sessions across ten trials are used for the mean population vector construction; subsequently, the relative change in PVD to ITI in each ensemble is calculated.

When the rotation of population vector between CS or US and ITI-1 sessions in CS- and US-responsive ensembles was calculated, we used the 1 s-averaged z-scores in CS- and US-responsive subpopulations.

### Immunohistochemistry and microscopy

Immunohistochemistry was conducted as described previously[32]. Mice were deeply anesthetized with the mixed anesthesia solution described above and perfused transcardially with 4% paraformaldehyde in PBS (pH 7.4). Brains were removed and further post-fixed by immersion in 4% PFA in PBS for 24 h at 4 °C. Each brain was equilibrated in 25% sucrose in PBS for 2 days and then frozen in dry ice powder. Fifty μm coronal sections were cut on a cryostat and stored at −20 °C in cryoprotectant solution (25% glycerol, 30% ethylene glycol, 45% PBS) until further use. For immunostaining, sections were transferred to 12-well cell culture plates (Corning, USA) containing Tris-buffered saline TBS-T buffer (with 0.2% Triton X-100, 0.05% Tween-20).

For EYFP or G-CaMP and/or RGS-14 detection, after washing with TBS-T buffer, the floating sections were treated with blocking buffer (5% normal donkey serum [S30, Chemicon, USA] in TBS-T) at room temperature for 1 h. Primary antibody incubations were performed in blocking buffer containing rabbit anti-GFP (1:500, A11122; Molecular Probes, USA) and/or mouse anti-RGS-14 (1:500, N133/21; NeuroMab, USA) antibodies at 4 °C for 1–2 days. After three 20 min washes with TBS-T, the sections were incubated with donkey anti-rabbit IgG-AlexaFluor 488 (1:500, A21206; Molecular Probes) and/or donkey anti-mouse IgG-AlexaFluor 546 secondary antibodies (1:500, A11036; Molecular Probes) in the blocking buffer at room temperature (RT) for 3 h.

For tdTomato detection, after washing with TBS-T buffer, floating sections were treated with blocking buffer (5% normal goat serum [S1000, Vector Laboratories, USA] in TBS-T) at RT for 1 h. Incubation with primary antibodies was performed in blocking buffer containing rabbit anti-DsRed (1:1000, 632496; Clonetech-Takara Bio, Japan) antibody at 4 °C for 1–2 days. After three 20 min washes in TBS-T, sections were incubated with goat anti-rabbit IgG-AlexaFluor 546 secondary antibodies (1:300, A11035; Molecular Probes) in blocking buffer at RT for 3 h. Sections were treated with DAPI (1 μg/mL, 10236276001; Roche Diagnostics, Switzerland) and then washed with TBS three times (20 min/wash). The sections were mounted on slide glass with ProLong Gold antifade reagent (Invitrogen, USA). Images were acquired using a Keyence microscope (BIO-REVO, KEYENCE, Japan) with a Plan-Apochromat 4× or 20× objective lens.

### Statistics

Data are presented as means ± s.e.m. unless specified otherwise. Box plots represent median, first, and third quantiles, and their whiskers show minimum and maximum values. In box plots, outlier values are not shown for clarity of presentation, but all data points and animals were included in statistical analyses. Statistical analyses were performed using Excel (Microsoft) with Statcel4 (OMS, Japan) and MATLAB (Mathworks, USA) as described previously[32]. Comparisons of data between two groups were analyzed with a two-tailed Student's $t$ test, Wilcoxon rank sum test, Wilcoxon signed-rank test, or Brunner-Munzel Test based on the distribution and "n" size of the data. Correlation was analyzed with a Pearson correlation coefficient test. Multiple-group comparisons were conducted using two-way analysis of variance (ANOVA) with a two-tailed post-hoc Tukey–Kramer multiple comparisons test when significant main effects were detected. Adjustments were made for Tukey–Kramer's multiple comparison test. Quantitative data are expressed as means ± SEM.

### Reporting summary

Further information on research design is available in the Nature Portfolio Reporting Summary linked to this article.

## Data availability

The data are available from the corresponding author upon request. The datasets supporting this study will be deposited to a public repository when the ongoing studies using the same dataset are published. Source data are provided with this paper.

## Code availability

All codes for the manuscript are available at https://github.com/IdlingBrainUT/Nomoto2022_NatureCommunications and https://doi.org/10.5281/zenodo.7293824.

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

## Acknowledgements

We are grateful to K. Deisseroth (Stanford University) for providing eArchT3.0-EYFP cDNA; S. Tonegawa (MIT) and S. Itohara (RIKEN) for providing floxed-NR1 transgenic mice; T. Fukai (Okinawa Institute of Science and Technology) and T. Haga (RIKEN) for mathematical analysis; T. Takekawa (Kogakuin University) for early access to the HOTARU detection system; M. Ito and N. Takino (Jichi Medical University) for production of the AAV vector. We thank alumni and current members of the Inokuchi laboratory for discussion. This work was supported by JSPS KAKENHI (grant number: JP18H05213), the Core Research for Evolutional Science and Technology (CREST) program (JPMJCR13W1) of the Japan Science and Technology Agency (JST), a Grant-in-Aid for Scientific Research on Innovative Areas "Memory dynamism" (JP25115002) from MEXT support, the Takeda Science Foundation to K.I., and the Grant-in-Aid for JSPS KAKENHI Scientific Research(B) (20H03554), Challenging Research (Exploratory) (17K19445), THE HOKURIKU BANK Grant-in-Aid for Young Scientists, the FIRSTBANK OF TOYAMA SCHOLARSHIP FOUNDATION RESERCH GRANT, the Takeda Science Foundation, the Tamura Science and Technology Foundation, and the Narishige Neuroscience Research Foundation support to M.N.

## Author contributions

Conceptualization: M.N., K.I. Formal analysis: M.N., E.M., R.O.S., S.O., M.N. Funding acquisition: M.N., K.I. Investigation: M.N., E.M. Methodology: M.N., K.I. Project administration: K.I. Software: M.N. A.A.V. vector production: S.M. Supervision: K.I. Visualization: M.N. Writing – original draft: M.N. Writing – review & editing: M.N., K.I.

## Competing interests

The authors declare no competing interests.
