## [Peer Review File · Nature Communications]

Hippocampus as a sorter and reverberatory integrator of sensory inputsREVIEWER COMMENTS

Reviewer #1 (Remarks to the Author):

Nomoto et al. showed that CA3-NMDA receptor (NR1) deficit mice and mice with CA3 ontogenetically silenced showed impaired long-term cued-fear memory. Further, calcium events of conditioning-stimuli (CS) and test-CS responsible cells in CA1 are more activated after presenting unconditioned stimuli (US) during conditioning sessions in wild-type than CA3 NR deficit mice. After the US presentation, the triple-wise synchronous activity of CS, US, and test-CS cells was higher in wild-type than CA3 NR deficit mice. The incidence of triple-wise synchronous activity was positively correlated with freezing behavior. The experiments and analysis are sound, and the results are novel and important. Thus, I support, in principle, the publication of this manuscript. I have several comments.

1)The authors repeatedly use the word “reverberatory” in the manuscript, including in the abstract. The activity of CS, US, and future test-CS cells is enhanced during inter-trial intervals after the US presentation. Still, I did not understand in which sense the authors called this activity “reverberatory.” Please clarify or rephrase it.

2)The authors use the word “connectivity”; however, they measure the synchrony of cells using a relatively wide (500 ms) time bin. Therefore, I suggest using “synchrony” instead of “connectivity.”

3)Traditional engram cell theory postulated that cells activated during memory encoding are reactivated during memory retrieval. However, the authors found that a significant fraction of test-CS cells and CS or US cells do NOT overlap, and thus their findings are, at first glance, contradict the memory engram cell hypothesis. Therefore, I suggest discussing the relationship between engram cell theory and their results.

4)In cued-fear memory experiments, a recent study reported the across-regional co-activation of neuronal ensembles in the hippocampus, amygdala, and prefrontal cortex during conditioning, memory consolidation, and memory retrieval (Miyawaki and Mizuseki, Nature Commun 2022). Comparing their results and Nomoto et al.’s results may highlight the importance and novelty of this manuscript.

5)Data and code availability. The manuscript, including the Datasheet, contains only highly processed data. Not only highly processed data but also raw data should be declared to be available upon request or should be deposited as an open-access database.

Reviewer #2 (Remarks to the Author):

The theoretical models of hippocampal area CA3 postulate that reverberatory activity caused by massive recurrent networks among CA3 cells could serve as an associator or integrator of multiple sensory inputs. However, Lovett-Barron et al (2014) suggested that the temporoammonic pathway from entorhinal cortex to CA1 prevents association between condition stimuli (CS) and unconditioned stimuli (US) during CS and US representation. Here authors hypothesized that CS and US association occurs after the termination of sensory inputs, and tested whether and how reverberatory activity contributing to the CS-US association during Light-cued fear conditioning (LFC) paradigm in mice, using in vivo Ca imaging. They used CA3-NR1 KO mice (or optogenetic silencing of CA3) to disturb the CA3 function.

After confirming that long-term (24-hr) cued fear conditioning depends on CA3 (Fig 1), they monitored in vivo Ca dynamics of CA1 cells and CA3 cells, separately. During LFC, the same CA cells were tracked across LFC sessions. While CS-responsive cells, US-responsive cells and Test-CS cell (that appeared during test phase), were newly emerged in CA1 (Fig 2) and CA3 (Fig 5), CA3 NR1 KO displayed no effect on the ensemble responses to CS and US inputs for each area. Interestingly, CS cell activity during ITI-1 continued as reverberatory activity in CA1 (Fig 3a) and CA3 (Fig 6a), which are diminished in the CA3 NR1 KO mice. CA3 NR-dependent reverberatory activity in CA1 and CA3 was also observed in the head-fixed conditions (Fig 4). Further, Test-CS cell activity emerged during ITI-1 in CA1 (Fig 3c) and during ITI-2 in CA3 (Fig 6c), which were not obvious in the KO mice. Moreover, connectivity (the measure of integration) among non-shared cells was increased during ITI-1 in CA1 (Fig 3e) but not in CA3 (Fig 6e), and heightened connectivity in CA1 was impaired in the CA3 KO mice (Fig 3e). These findings suggested that CA3 acts as a reverberator during ITI and CA1 acts as both a reverberator and an integrator (for memory association) of the sensory inputs. Finally, the authors tested whether CA3 reverberatory activity is necessary for association of CS and US. When CA3 neuronal activity was optogenetically silenced during ITI, mice were impaired in cued-fear recall, but not contextual fear memory (Fig 7).

Experimental procedures are technically sound and largely consistent each other. I have only a few issues that might improve the manuscript.

Major:

1. On page 8, line 179-181 for In Fig 6g,h, it said, the CS ensemble of the CA3 KO mice exhibited significant rotation from CS to ITI. However, Fig 6g,h does not show significance. I felt mismatch between the main text and Figure data. Please clarify.
2. The unit of connectivity in Fig 6e is 10^{-2} to 10^{-4} . Extremely low. Is this mistypo? If this is not simple mistypo, the reason why no difference between the CA3 mutants and controls is due to the floor effect.
3. In Fig 6g,h, the mutants' animal number is only 4, while the controls are 11, which made the mutants' error bars much bigger. In contrast, in Fig 3g,h, the number of KO is 11 while controls' number is 6. Why so different and actually use oppositely? Unless increasing the KO mutants' number in Fig 6g,h, it is premature to conclude that CA3 acts as reverberator only (not as integrator). Please justify the statistical

sample size for both Fig 3g,h and 6g,h, and, if necessary, please increase the KO mouse number in Fig 6g,h.

4. They found that contextual fear memory recall is intact in CA3 manipulation. It is tempting to hear why contextual recall is intact compared to cued recall based on the current study results, while some previous studies also reach the same conclusion.

Minor:

1. Text of Abstract is awkward, for example, “occur” appears twice line 24 in Page 2. Please revise the Abstract.

2. In Extended Fig 6, why are there so many active purple cells in the Test-CS cells during Recall, despite “no episodic recall”?

3. In the raster plots of Figs 3a-c, 4c-f, and 6a-c, no genotypes are mentioned. I guess the left plots are all controls and right plots are all KOs.

Response to Reviewers' comments:

We thank the reviewers for the helpful and insightful comments and useful suggestions. We have responded to each of the reviewers concerns and have modified the manuscript accordingly (all changes are highlighted in red). We have also reformatted the manuscript to comply with the style of Nature Communications and have updated some figures because of additional experiments and analyses.

Reviewer #1 (Remarks to the Author):

Nomoto et al. showed that CA3-NMDA receptor (NR1) deficit mice and mice with CA3 ontogenetically silenced showed impaired long-term cued-fear memory. Further, calcium events of conditioning-stimuli (CS) and test-CS responsible cells in CA1 are more activated after presenting unconditioned stimuli (US) during conditioning sessions in wild-type than CA3 NR deficit mice. After the US presentation, the triple-wise synchronous activity of CS, US, and test-CS cells was higher in wild-type than CA3 NR deficit mice. The incidence of triple-wise synchronous activity was positively correlated with freezing behavior. The experiments and analysis are sound, and the results are novel and important. Thus, I support, in principle, the publication of this manuscript. I have several comments.

Response: We are grateful to the reviewer for the helpful comments and useful suggestions, which have helped us to improve the manuscript.

1)The authors repeatedly use the word “reverberatory” in the manuscript, including in the abstract. The activity of CS, US, and future test-CS cells is enhanced during inter-trial intervals after the US presentation. Still, I did not understand in which sense the authors called this activity “reverberatory.” Please clarify or rephrase it.

Response

We respect the reviewer's comment. We have used the term, reverberatory activity, as a trace activity of cells that is evoked by sensory inputs and transiently lasts after the sensory inputs ceased.

The term, reverberatory activity, was initially conceptualized by Hebb (Chapter 4: The First Stage of Perception: Growth of the Assembly, Organization of Behavior, 1949). Recent review articles also supported the concept of reverberatory activity (Johnson et al., 2009, Yuste. 2015). Thus, the term reverberatory activity has been

popularly used in the neuroscience field, but its function and mechanism were not clear until our current study. To clarify the meaning of reverberatory activity, we have revised introduction section (please see introduction section, page 3-4).

Revised introduction section

We defined the reverberatory activity as a trace activity of cells that is evoked by sensory inputs and transiently lasts after the sensory inputs ceased.

References (for reverberatory activity, cited in the manuscript)

Hebb, D. O. The organization of behavior: a neuropsychological theory. (J. Wiley; Chapman & Hall, 1949).

Johnson, L. R., Ledoux, J. E. & Doyere, V. Hebbian reverberations in emotional memory micro circuits. Front Neurosci 3, 198-205 (2009).

Yuste, R. From the neuron doctrine to neural networks. Nat Rev Neurosci 16, 487-497 (2015).

2)The authors use the word “connectivity”; however, they measure the synchrony of cells using a relatively wide (500 ms) time bin. Therefore, I suggest using “synchrony” instead of “connectivity.”

Response

In accordance with the reviewer’s suggestion, we have changed terms "connectivity" and “connection” to "synchrony" throughout the manuscript.

3)Traditional engram cell theory postulated that cells activated during memory encoding are reactivated during memory retrieval. However, the authors found that a significant fraction of test-CS cells and CS or US cells do NOT overlap, and thus their findings are, at first glance, contradict the memory engram cell hypothesis. Therefore, I suggest discussing the relationship between engram cell theory and their results.

Response

Thank you for this insightful suggestion. Test-CS cells showed activity not only during CS presentation in test session but also during reverberatory phase (ITI-1) of conditioning (Fig. 3c). Given that conditioning was consisted of CS, US, and ITI phases, Test-CS cells behave like the theorized engram cells, in which cells activated during conditioning (ITI) are re-activated in recall. Thus, Test-CS cell population serves as engram that drive the

episodic recall of cued-fear memory. Please note that, in this study, CS cells are defined as cells that exhibited higher responses during CS period, which did not include the ITI period. This results in less overlapping between test-CS cells and CS or US cells. We suggest that Test-CS cells were recruited during ITI (reverberation) by synchronized activities between CS and US cells (Supplementary Fig. 6). To make this point clear, we have revised the discussion section (please see discussion section, page 10).

Revised discussion section

The significant correlation between relative degree of animal freezing and triple synchrony of non-shared cells during ITI strongly suggests that, by synchronized activity, CS and US cells recruit and instruct newly generated Test-CS cells by synchronizing CS and US information (Supplementary Fig. 6). Thus, Test-CS cell population serves as engram that drive the episodic recall of cued-fear memory.

4)In cued-fear memory experiments, a recent study reported the across-regional co-activation of neuronal ensembles in the hippocampus, amygdala, and prefrontal cortex during conditioning, memory consolidation, and memory retrieval (Miyawaki and Mizuseki, Nature Commun 2022). Comparing their results and Nomoto et al.'s results may highlight the importance and novelty of this manuscript.

Response

We thank the reviewer for this insightful suggestion. By comparing the Miyawaki and Mizuseki's finding, we have revised manuscript to discuss and highlight the importance and novelty of our manuscript (please see discussion section, page 11).

Revised discussion section

Recently, Miyawaki and Mizuseki have reported that the inter-regional coactivation of cell ensemble in the hippocampus, amygdala, and prefrontal cortex occurs during the cued-fear memory conditioning, consolidation, and retrieval⁴³. The inter-regional coactivation may contribute to the communication between the hippocampus and the amygdala, in which the episodic portion of the CS-US information is transported from the hippocampus to the amygdala.

Added new reference

Miyawaki, H. & Mizuseki, K. De novo inter-regional coactivations of preconfigured local ensembles support memory. Nat Commun 13, 1272 (2022).

5) Data and code availability. The manuscript, including the Datasheet, contains only highly processed data. Not only highly processed data but also raw data should be declared to be available upon request or should be deposited as an open-access database.

We have deposited the MATLAB codes used in this study to our laboratory Github repository (https://github.com/IdlingBrainUT/Nomoto2022_NatureCommunications). The data are available from the corresponding authors upon request. Source data are provided with this paper.

We have revised the statement for data and code availability in the method section (please see method section, page 43).

Revised method section

Data availability

The data are available from the corresponding authors upon request. Source data are provided with this paper.

Code availability

All codes for the manuscript are available at https://github.com/IdlingBrainUT/Nomoto2022_NatureCommunications

Reviewer #2 (Remarks to the Author):

The theoretical models of hippocampal area CA3 postulate that reverberatory activity caused by massive recurrent networks among CA3 cells could serve as an associator or integrator of multiple sensory inputs. However, Lovett-Barron et al (2014) suggested that the temporoammonic pathway from entorhinal cortex to CA1 prevents association between condition stimuli (CS) and unconditioned stimuli (US) during CS and US representation. Here authors hypothesized that CS and US association occurs after the termination of sensory inputs, and tested whether and how reverberatory activity contributing to the CS-US association during Light-cued fear conditioning (LFC) paradigm in mice, using in vivo Ca imaging. They used CA3-NR1 KO mice (or optogenetic silencing of CA3) to disturb the CA3 function.

After confirming that long-term (24-hr) cued fear conditioning depends on CA3 (Fig 1), they monitored in vivo Ca dynamics of CA1 cells and CA3 cells, separately. During LFC,

the same CA cells were tracked across LFC sessions. While CS-responsive cells, US-responsive cells and Test-CS cell (that appeared during test phase), were newly emerged in CA1 (Fig 2) and CA3 (Fig 5), CA3 NR1 KO displayed no effect on the ensemble responses to CS and US inputs for each area. Interestingly, CS cell activity during ITI-1 continued as reverberatory activity in CA1 (Fig 3a) and CA3 (Fig 6a), which are diminished in the CA3 NR1 KO mice. CA3 NR-dependent reverberatory activity in CA1 and CA3 was also observed in the head-fixed conditions (Fig 4). Further, Test-CS cell activity emerged during ITI-1 in CA1 (Fig 3c) and during ITI-2 in CA3 (Fig 6c), which were not obvious in the KO mice. Moreover, connectivity (the measure of integration) among non-shared cells was increased during ITI-1 in CA1 (Fig 3e) but not in CA3 (Fig 6e), and heightened connectivity in CA1 was impaired in the CA3 KO mice (Fig 3e). These findings suggested that CA3 acts as a reverberator during ITI and CA1 acts as both a reverberator and an integrator (for memory association) of the sensory inputs. Finally, the authors tested whether CA3 reverberatory activity is necessary for association of CS and US. When CA3 neuronal activity was optogenetically silenced during ITI, mice were impaired in cued-fear recall, but not contextual fear memory (Fig 7).

Experimental procedures are technically sound and largely consistent each other. I have only a few issues that might improve the manuscript.

We are grateful to reviewer for critical comments and useful suggestions that have helped us to improve our manuscript.

Major:

1. On page 8, line 179-181 for In Fig 6g,h, it said, the CS ensemble of the CA3 KO mice exhibited significant rotation from CS to ITI. However, Fig 6g,h does not show significance. I felt mismatch between the main text and Figure data. Please clarify.

Response

By verifying the data used for first submission, we confirmed that the CS ensemble of the CA3-NR1 KO mice exhibited modest but statistically significant rotation from CS to ITI ($p = 0.048$). However, according to the reviewer's comment (Reviewer 2, point 3), we have performed an additional experiment to increase the number of CA3-NR1 KO animals up to 9 mice in the CA3 imaging experiment. We have incorporated these new data and re-analyzed. Statistical analysis using the incorporated data showed that there were significant differences between control and CA3-NR1 KO animals both in Mahalanobis PVD and ensemble rotation in CS ensemble. These differences were not

observed in US ensemble (revised Fig 6i, j). We have now revised the manuscript by incorporating these data (please see page 8-9 and Fig. 6i, j).

Revised Fig. 6

i, j Mahalanobis PVD and rotation (**i**) between CS and ITI-1 sessions in the CS-responsive ensemble, and (**j**) between US and ITI-1 sessions in the US-responsive ensemble.

2. The unit of connectivity in Fig 6e is 10^{-2} to 10^{-4} . Extremely low. Is this mistypo? If this is not simple mistypo, the reason why no difference between the CA3 mutants and controls is due to the floor effect.

Response

We respect the reviewer’s critical comment. During the course of additional experiment and re-analyses, we found a mislabeling in the title of panels e and f of the initial submission; correctly panel e and f represents shared and non-shared cells, respectively. We have confirmed that the source data and statistical summary in the initial submission were correct.

The unit of connectivity in Fig 3e, 3f, 6e, and 6f is correct and not mistypo. It was due to the extremely large denominators. We calculated this normalized index as follows. In step 1, after the binarization of z-scored calcium activity, the number of total triple-wise connectivity was counted. In step 2, to normalize connectivity among animals, this total triple-wise connectivity was divided by maximal combination ($\#$ of CS cells \times $\#$ of US cells \times $\#$ of Test-CS cells = huge number). In step 3, to normalize the difference of the number of sampling frames in behavioral sessions, the value in step 2 was further divided by the number of frames of each behavioral session. These calculation steps are required to compare the unit of connectivity index across behavioral sessions and among animals.

In the revised manuscript, we have calculated “synchrony rate” in addition to the

synchrony index. The synchrony rate (Hz) is an indicator of the frequency of synchronous events during ITI-1 (Fig. 6g in the revised manuscript). (Please note that, in accordance with the reviewer's suggestion, comment 2 of reviewer 1, we have changed terms "connectivity" and "connection" to "synchrony" in the revised manuscript.)

The mean rates were 0.34Hz (CA1 of control mice), 0.33Hz (CA1 of CA3-NR1 KO mice), 0.11Hz (CA3 of control mice), and 0.13Hz (CA3 of CA3-NR1 KO mice). There was significant difference of synchrony rate between CA1 and CA3, and the synchrony rates were not extremely low. Please also see Supplementary Data for detailed statistics.

Regarding floor effect: We have added a graph showing synchrony index during ITI-1 in the revised manuscript (Fig. 6h). We observed significant differences in synchrony rate (Fig. 6g) and synchrony index (Fig. 6h) between CA1 and CA3, which indicates that no difference observed between control and CA3 NR-KO was not due to the floor effect.

These results support our conclusion that the CA1 acts as reverberator and integrator (because of the higher synchrony rate than CA3) while CA3 acts as reverberator only.

We have now incorporated these new data into the revised Figure 6 (please see page 8-9 and 6g, h).

Revised Fig. 6

g Synchrony rate between CA1 and CA3 and genotypes. **h** Synchrony index during ITI-1 in CA1 and CA3 and genotypes.

3. In Fig 6g,h, the mutants' animal number is only 4, while the controls are 11, which made the mutants' error bars much bigger. In contrast, in Fig 3g,h, the number of KO is 11 while controls' number is 6. Why so different and actually use oppositely? Unless increasing the KO mutants' number in Fig 6g,h, it is premature to conclude that CA3 acts

as reverberator only (not as integrator). Please justify the statistical sample size for both Fig 3g,h and 6g,h, and, if necessary, please increase the KO mouse number in Fig 6g,h.

Response

We respect the reviewer's critical comments. According to the reviewer's suggestions, we have performed an additional experiment to increase the number of KO animals up to 9 mice in CA3 imaging experiment. Statistical analyses using the incorporated data showed that there were significant differences between control and KO animals both in Mahalanobis PVD and ensemble rotation in CS, but not US, ensemble in CA3 (**revised Fig. 6i, j**). Based on the incorporated and re-analyzed data of ensemble activity (**revised Fig. 6a-c**), triple-wise synchrony (**revised Fig. 6e, f, g**), synchrony rate (**revised Fig. 6g**), and the data of head-fixed imaging experiment (**Fig. 4**), we conclude that CA3 acts as reverberator only (not as integrator).

We have revised the Supplementary Fig. 6 to highlight the higher CA1 synchrony index in control mice compared to CA3-NR1 KO in spite of the comparable CA1 synchrony rate between genotypes.

We have incorporated these new data and revised the manuscript and figures (**please see page 8-10; see also revised Figure 5 and 6, and Supplementary Fig 6**).

Revised Fig. 5

a Left, experimental design. Right, coronal section of the hippocampus with G-CaMP-expressed cells (green) in CA3 and immunostained with anti-RGS-14 (red).

b Imaging data analysis scheme.

c Columns comparing the number of detected cells during CA3 imaging in control and CA3-NR1 KO mice.

d Columns comparing percentiles of ensemble sizes in CS-, US-, and Test-CS-responsive subpopulations between control and KO mice.

e Box plots comparing mean z-scores of long-term memory test sessions between genotypes in CS-, US-, and Test-CS-responsive subpopulations.

f Venn diagrams comparing and illustrating the overlapping and size of each ensemble in CA3. Columns compare the percentiles of overlapping ensemble sizes between CS-, US-, and Test-CS-responsive subpopulations between control and KO mice.

Revised Fig. 6

a-c Left, venn diagrams representing (a) CS-, (b) US-, and (c) Test-CS-responsive ensembles. Peri-event raster plots during the training session in each subpopulation of littermate control and KO mice. Middle, averaged z-score plots over ten CS-US pairings in each subpopulation. Right, box plots comparing mean z-scores between genotypes in each session.

d Left, representative binarized raster plots of Ca²⁺ activity across ten ITI-1 sessions in control animals. Binarized synchrony data is used for calculation of synchrony rate.

e, f Box plots comparing mean synchrony between genotypes in each session.

g Synchrony rate between CA1 and CA3 and genotypes.

h Synchrony index during ITI-1 in CA1 and CA3 and genotypes.

i, j Mahalanobis PVD and rotation (i) between CS and ITI-1 sessions in the CS-responsive ensemble, and (j) between US and ITI-1 sessions in the US-responsive ensemble.

Revised results section

CA3 NRs are important for reverberation of CS stimuli in freely moving condition

We also examined CA3 dynamics during the LFC task in freely moving conditions (Fig. 5a,b). There were no significant differences in the cell ensemble structure or in the Ca²⁺ activities during test sessions between CA3-NR1 KO mice and littermate controls (Fig. 5c-f). **Similar to the CA1, US input partially inhibited the Ca²⁺ activities of CS- but not Test-CS-responsive cells.** The CS-responsive cells in CA3-NR1 KO mice exhibited significantly lower Ca²⁺ activities **during the 9 s of the ITI-1 (6 to 14 s of 15 s in ITI-1, later 9s)** than the control (Fig. 6a). **CS-, US-, and Test-CS-responsive cells in CA3-NR1 KO exhibited significantly lower Ca²⁺ activities during the ITI-2 sessions than the control (Fig. 6a-c).** Triple **synchrony** was comparable between CA3-NR1 KO and littermate control mice for the duration of **LFC training** sessions (Fig. 6d-f).

We further compared the synchrony rate (Hz) in ITI-1 that is an indicator of the frequency of synchronous events (Fig. 6g). Two-way ANOVA for the synchrony rate in ITI-1 revealed a significant effect of hippocampal region, but not genotype and hippocampal region vs. genotype interaction (Fig. 6g). Two-way ANOVA for the synchrony index in ITI-1 revealed a significant effect of interaction between hippocampal region vs. genotype (Fig. 6h). The CA1 synchrony index during ITI-1 in littermate control was higher than the other groups (Fig. 6h). Thus, the deficit of CA3 NRs reduces the CA1 synchrony to the same level as CA3.

The ensemble representation analyses revealed that the CS ensemble representations, but not US, were more stable in littermate control mice than in CA3-NR1 KO mice across sessions, as demonstrated by both small PVD and ensemble rotation changes from the stimulus to the ITI session (Fig. 6i,j). These hippocampal CA3 and CA1 imaging results suggested that after termination of sensory stimuli, the CA3-CA1 pathway acts as a reverberatory network of episodes in a CA3 NR-dependent manner.

Revised discussion section

We detected time-limited and CA3 NR-dependent reverberatory activities that lead to synchronized activity among cell ensembles in the CA1. **Our results suggest that the CA1 integrates episodes through the synchronized activity during the reverberation phase.** The CA3 to CA1 network functions as a reverberatory and associative system of stimuli, in which the CA3 acts as a reverberator and the CA1 functions as both a reverberator and

an integrator of episodes.

Revised Supplementary Fig. 6

Revised Supplementary Fig. 6 legend

a Left, Venn diagrams showing CS-, US-, and Test-CS-responsive cell ensembles in CA1 of normal mice, in which CA3-dependent reverberation occurs normally. Right, raster plots of CA1 subpopulations and with the timeline of the cued-fear memory paradigm. During CS and US inputs during training, CS and US information are separately encoded in CS- and US-responsive cell populations, respectively. During reverberation in training, co-activity of CS- and US-responsive cells recruits and instructs Test-CS-responsive cells in the CS-US episode. During recall, Test-CS-responsive cells drive the episodic recall of cued-fear memory.

b Left, Venn diagrams of CS-, US-, and Test-CS-responsive ensembles in loss-of-function (CA3-NR1 KO and CA3 silencing during ITI). Right, during CS and US input in training, CS and US information are separately and normally encoded. However, without reverberation in training, the low co-activity of CS- and US-responsive cells fails to recruit and instruct Test-CS-responsive cells in the episodic relation between CS and US. Thus, during recall, Test-CS-responsive cells fail to drive cued-fear memory recall. **Note**

that synchrony rate in reverberation is comparable between genotypes. In contrast, synchrony index, which reflects the number of CS and Test-CS cells contributing to the synchrony events, is lower in CA3-NR1 KO mice. Filled circles with color indicate activated cells in each behavioral session. Arrows indicate the direction of information flow. Light bulb, light CS; yellow bar, moment-occurring synchrony among CS-, US-, and Test-CS cell ensembles.

The reason why different number of animals were used: In this study, after the finding of CA3 involvement in the cued-fear memory recall, we first conducted the CA3 imaging with control and CA3-NR1 KO animals. We observed that CA3-NR1 KO, but not control, mice implanted with prism GRIN lens into CA3 did not recover normally from the implantation operation in many cases. The KO mice that did not recover normally died within 3 days after the lens implantation operation. On the other hand, recovered KO animals normally grew and were healthy throughout the experiment. Previous report showed that CA3-NR1 KO mice show the susceptibility to kainic acid (KA)-induced seizures (Jinde et al. Eur J Neurosci 30, 1036-1055, 2009). We considered that the attachment and implantation of prism GRIN lens into CA3 may transiently induce the slight mechanical stretch that activates cells in CA3 during first recovery period after surgery, leading to weak seizure in CA3 KO animals. Given that the recovered CA3-NR1 KO animals normally grew and were healthy afterward and that the responses of the CA3 GCaMP to sensory inputs in CA3-NR1 KO animals were comparable to control animals, we believe that the CA3 calcium imaging data in healthy CA3-NR1 KO are appropriate for further analyses to reach our conclusions.

Subsequently, we moved to the CA1 imaging experiment and operated a greater number of the CA3-NR1 KO animals than control animals. However, most of the CA3-NR1 KO animals (CA1-implanted) normally recovered, so eventually we obtained data from 11 KO animals.

Reference

Jinde, S. et al. Lack of kainic acid-induced gamma oscillations predicts subsequent CA1 excitotoxic cell death. Eur J Neurosci 30, 1036-1055 (2009).

The sample size of this study was based on works in previous publications (Ohkawa et al. Cell Reports 11, 261-269, 2015; Nomoto et al. Nature Communications 7, 12319, 2016; Yokose et al. Science 355, 398-403, 2017; Ghandour, K. et al Nature Communications 10,

2637, 2019). We believe that the sample size used in this study, after the addition experiment, is sufficient to support our conclusions.

Added reference

Ghandour, K. et al. Orchestrated ensemble activities constitute a hippocampal memory engram. *Nat Commun* 10, 2637 (2019).

4. They found that contextual fear memory recall is intact in CA3 manipulation. It is tempting to hear why contextual recall is intact compared to cued recall based on the current study results, while some previous studies also reach the same conclusion.

Response

In our experimental protocol, mice were first habituated to the context for four consecutive days, which generates contextual memory. Mice were then subjected to the conditioning, followed by LFC test and then CFC test. We think that without the contextual habituation CA3 manipulation results in impairment in CFC memory as well. This point was already discussed in the initial manuscript. (please see discussion section, page 11).

Related discussion section

Consistent with previous reports²⁻⁴, NR deficiency in the CA3 or CA3 silencing during reverberation did not impair contextual fear memories after pre-contextual habituation (Fig. 1e, 7e, and Supplementary Fig. 1). These findings suggest that reverberation is required for association of novel episodes but not for association with pre-existing memories.

Minor:

1. Text of Abstract is awkward, for example, “occur” appears twice line 24 in Page 2. Please revise the Abstract.

Response

Thank you for your helpful suggestion. In accordance with the reviewer’s suggestion, we have revised abstract (please see abstract, page 2).

2. In Extended Fig 6, why are there so many active purple cells in the Test-CS cells during Recall, despite “no episodic recall”?

Response

The number of Test-CS cells in CA1 was comparable between genotypes (Fig. 2d). Thus, the number of purple cells (just respond to test-CS input) and green cells (harboring memory) is same (please see Fig. 2d).

Fig. 2

d Columns comparing percentile of ensemble size in CS-, US-, and Test-CS-responsive subpopulations between control and KO mice.

3. In the raster plots of Figs 3a-c, 4c-f, and 6a-c, no genotypes are mentioned. I guess the left plots are all controls and right plots are all KOs.

Response

Thank you for your helpful suggestion. Left plots are all controls and right plots are all KOs. Genotypes are mentioned in these figures in the revision (please see revised figures 3a-c, 4c-f, and 6a-c).

Revised Fig. 3a-c

a-c Left, Venn diagrams representing (a) CS-, (b) US-, and (c) Test-CS-responsive ensembles. Peri-event raster plots during the training session in each subpopulation of littermate control and KO mice.

Revised Fig. 4c-f

c-f Left, peri-event raster plots during single **(c)** CS presentation session in CA1, **(d)** US presentation session in CA1, **(e)** CS presentation session in CA3, and **(f)** US presentation session in CA3 of (left) control and (right) KO mice.

Revised Fig. 6a-c

a-c Left, venn diagrams representing **(a)** CS-, **(b)** US-, and **(c)** Test-CS-responsive ensembles. Peri-event raster plots during the training session in each subpopulation of littermate control and KO mice.

REVIEWERS' COMMENTS

Reviewer #1 (Remarks to the Author):

The authors have thoughtfully responded to my previous comments and the comments from the other reviewer. I think “percentile” in the legend of Fig 2d (line 331), 2f (line 336), 5d (line 401), 5f (line 406) should be replaced with “percent”. I have no other comments.

Reviewer #2 (Remarks to the Author):

In the revised manuscript, the authors fully responded to my comments. In particular, they conducted an additional experiment to increase the number of KO mice up to nine mice for the CA3 imaging. Consequently, their conclusion that CA3 acts as a reverberator, but not as integrator, has further been supported by the data analysis.

Response to Reviewers' comments:

We thank the reviewers for the helpful and insightful comments and useful suggestions. We have responded to the Reviewer #1 suggestion and have modified the manuscript accordingly (all changes are highlighted in red). We have also reformatted the manuscript to comply with the Author Checklist of Nature Communications.

Reviewer #1 (Remarks to the Author):

The authors have thoughtfully responded to my previous comments and the comments from the other reviewer. I think “percentile” in the legend of Fig 2d (line 331), 2f (line 336), 5d (line 401), 5f (line 406) should be replaced with “percent”. I have no other comments.

Response: We are grateful to the reviewer for the helpful comments and useful suggestions, which have helped us to improve the manuscript. In accordance with the reviewer’s critical suggestion, we have changed terms "percentile" to "percent" throughout the manuscript (please see revised figure legend for Fig 2d (line 849), 2f (line 859), 5d (line 932) and 5f (line 940)).

(Please note that, in accordance with the Author Checklist, we have changed the order of the sections in the manuscript, so the line numbers for figure legend have changed.)

Reviewer #2 (Remarks to the Author):

In the revised manuscript, the authors fully responded to my comments. In particular, they conducted an additional experiment to increase the number of KO mice up to nine mice for the CA3 imaging. Consequently, their conclusion that CA3 acts as a reverberator, but not as integrator, has further been supported by the data analysis.

Response: We are grateful to the reviewer for the helpful comments and useful suggestions, which have helped us to improve the manuscript.